# Pupil-linked arousal signals track the temporal organization of events in memory

David Clewett[1], Camille Gasser[2] & Lila Davachi [2,3✉]

Everyday life unfolds continuously, yet we tend to remember past experiences as discrete event sequences or episodes. Although this phenomenon has been well documented, the neuromechanisms that support the transformation of continuous experience into distinct and memorable episodes remain unknown. Here, we show that changes in context, or event boundaries, elicit a burst of autonomic arousal, as indexed by pupil dilation. Event boundaries also lead to the segmentation of adjacent episodes in later memory, evidenced by changes in memory for the temporal duration, order, and perceptual details of recent event sequences. These subjective and objective changes in temporal memory are also related to distinct temporal features of pupil dilations to boundaries as well as to the temporal stability of more prolonged pupil-linked arousal states. Collectively, our findings suggest that pupil measures reflect both stability and change in ongoing mental context representations, which in turn shape the temporal structure of memory.

[1] Department of Psychology, New York University, New York, NY, USA. [2] Department of Psychology, Columbia University, New York, NY, USA. [3] Nathan Kline Institute, Orangeburg, NY, USA. ✉email: ld24@columbia.edu

As our lives unfold, we encounter a constant stream of sensory information. But our memories do not strictly mirror the time and tide of experience. Instead, we remember the past as a series of discrete and meaningful episodes, or events. Furthermore, memory for the temporal duration of these episodes is subjective and often prone to distortion. For example, even if two experiences had occurred across the same objective amount of time, memory for their duration can be modulated by the content and structure of their constituent events[1–5]. While there has been intense interest in characterizing the factors that modulate this transformation from continuous experience into discrete episodic memories, little is known about the neural processes that support such memory-structuring. Thus, the aim of the present series of experiments is to address a critical question in learning and memory research: What brain mechanisms facilitate the creation of a new episode in episodic memory?

Increasing research suggests that contextual stability over time plays a key role in integrating sequential information into memorable events. For instance, remaining in the same spatial context for an extended period of time, such as cooking breakfast in your kitchen, may help to organize a sequence of actions, such as cracking eggs and then frying them, into a unified event representation of eating breakfast at home[6,7]. However, when the surrounding context changes, such as entering a new room or being interrupted by a phone call, people tend to perceive an event boundary that defines the end of the current event and the beginning of a new one[7,8]. Importantly, these event boundaries have a significant impact on how we remember experiences later on by promoting more separated memory representations[1,4,5,9–18]. Thus, temporal stability and change in an unfolding context, including fluctuations in an individual's surroundings or mental state, help to form a mental timeline of discrete episodic memories.

To date, the primary method of indexing the formation of discrete event memories has been to examine memory for the order and duration of sequential information (for a review, see ref. [10]). For instance, when individuals are presented with two items from a recent experience, memory for the order of those items is relatively impaired if there was an intervening context change compared to instances where the items were encountered in the same context[12–19]. Likewise, item pairs are also remembered as having occurred farther apart in time if they spanned an event boundary compared to items that had been encountered in the same context[1,4,18]. Event boundaries can also influence non-temporal aspects of episodic memory, such as enhancing associative memory for an item and any co-incident contextual information (e.g., background color[16,20]). The existing literature therefore presents a complex story, whereby temporal aspects of episodic memory integration are disrupted by changes in the surrounding context, while other elements of the new context present at event boundaries are enhanced in memory.

We propose that one solution to this puzzle may relate to fluctuations in physiological arousal over the course of experience—a notion inspired by evidence that spikes in arousal yield strikingly similar effects on episodic memory as do shifts in context. First, like event boundaries, emotional stimuli or acute stressors that induce arousal elicit exaggerated estimates of time duration[21,22]. Second, viewing highly arousing videos prior to navigation or sequence learning has been shown to impair temporal order memory for neutral events[23,24]. Third, emotionally arousing stimuli can also enhance local item-context source memory[25,26].

Importantly, fluctuations in arousal are induced by more than just emotion and stress. Many salient environmental changes, such as hearing an unexpected sound, can activate central arousal systems that regulate ongoing attention and memory processes across the brain[27–29]. Furthermore, arousal signals mediate some of the same cognitive processes that are thought to be triggered by event boundaries, including cognitive control, prediction errors, and attention re-orienting[7,18,29–31]. Of relevance to the current study, emerging evidence also suggests that arousal responses are sensitive to the structure of temporally extended experiences[32,33]. For instance, pupil dilation occurs when a highly organized and repeated sequence of auditory tones suddenly transitions to a randomized sequence of tones, but not during the opposite transition[33]. This suggests that pupil-linked arousal processes modulate and/or signal disruptions in an ongoing stable context in a manner consistent with the presence of event boundaries. Critically, however, it remains unclear whether these dynamic arousal responses also relate to event-structuring effects in subsequent memory. In light of this evidence, we hypothesize that arousal systems are ideally positioned to translate the temporal of structure of experience into temporally organized memories.

In the current series of experiments, we test this hypothesis by monitoring pupil size during a sequence-learning task and examining if event boundaries trigger momentary bursts of arousal to promote event separation in memory. Here, participants encode a series of 32 everyday objects displayed on a computer. To manipulate event structure during learning, we use an auditory context manipulation wherein a simple tone is played in participants' left or right ear before each item. This tone remains the same for eight successive items, and then switches to the other ear to create an auditory event boundary, thereby parsing the continuous 32-item sequence into four discrete subevents. After each sequence, we then query participants' memory for the temporal order and temporal distance between the studied item pairs. Unbeknownst to participants, these pairs always appeared the same objective distance apart during encoding. Critically, some item pairs were encountered in the same auditory event, whereas other item pairs spanned an event boundary. Based on prior work, we predict that these boundaries will lead to relatively larger retrospective estimates of temporal distance between item pairs and impaired temporal order memory for those pairs. Additionally, we predict that boundaries will enhance participants' auditory source memory for items appearing immediately after a tone switch (henceforth referred to as boundary items) relative to same-context items.

In Experiments 2 and 3, we also measure pupil diameter continuously throughout our sequence-learning task to test our key hypothesis that boundary-induced autonomic arousal responses relate to later memory separation effects[34,35]. Accumulated evidence in humans and animals shows that pupil diameter may be a reliable marker of cognitive processing[36–38]. Increasing work also suggests that a temporal principal component analysis (PCA) can be used to decompose pupil measures into dissociable features that reflect distinct cognitive processes[39–44]. For instance, one recent study used PCA decomposition to show that a specific sub-component of pupil dilation is triggered just prior to the onset of memory retrieval decisions, suggesting that this response might signal the anticipation of an impending decision/response[40]. Temporal PCA has also been combined with studies manipulating different lighting conditions to distinguish overlapping contributions of parasympathetic and sympathetic autonomic pathways to pupil dilation[42,43]. A PCA thereby holds important advantages over more conventional pupil-averaging analyses and offers a unique window into how different mental (e.g., anticipation, motor responses) and neural processes may be engaged by event boundaries to structure subsequent memory.

We find that distinct temporal characteristics of pupil responses to event boundaries lead to changes in subjective (time dilation) and objective (impaired recency discrimination) aspects of temporal memory. Trial-level analyses reveal that greater

**Fig. 1 Auditory event boundary paradigm.** Participants studied lists of 32 everyday objects and had to indicate whether each item would more likely be encountered in an indoor or outdoor setting. The surrounding context was manipulated by playing a simple tone in either participants' left or right ear prior to viewing each image, which indicated to participants which hand they should use to make their subsequent indoor/outdoor judgment. After eight successive items, the tone switched to the other ear and changed in pitch. These tone switches served as event boundaries, which parsed each continuous 32-item sequence into four discrete auditory events. After a short distractor task, participants performed three different memory tests in two separate blocks of trials: the first block included two different temporal memory tests and the second block included a source memory test. In the temporal memory block, participants first had to indicate which of the two presented items had appeared more recently in the prior sequence. Second, they had to rate the temporal distance between these items, ranging from 'very close' to 'very far' apart in the sequence. After being tested on item pairs, participants performed an auditory source memory test for all of the remaining items that were not shown during the temporal memory tests. In this source memory test, participants were shown individual items and had to indicate whether each item had been paired with a tone in their left ear or their right ear. The hand button-press icon was made by Freepik from www.flaticon.com.

variability in pupil diameter over more prolonged periods of time is also associated with these segmentation-like effects in memory. We conclude that dynamic fluctuations in pupil-linked arousal states may track multiple mental processes that promote the formation of discrete episodic memories.

## Results

**Experiment 1: Behavior.** We first examined how context shifts influenced response times for an indoor/outdoor judgment that participants made for each item (Fig. 1). As expected, participants were significantly slower to make judgments about objects appearing immediately after a tone switch compared to items appearing after a repeated, or same-context tone, $t(33) = 4.06$, $p < 0.001$, $d = 0.70$, [CI: 36.37, 109.61] (Fig. 2).

Next, we examined how boundaries modulated the temporal structure of memory. First, we found that boundaries elicited a subjective time expansion effect in memory, such that boundary-spanning item pairs were later remembered as having appeared farther apart in time than same-context pairs, despite the objective distance being identical, $t(33) = 2.44$, $p = 0.02$, $d = 0.42$, [CI: 0.015, 0.17] (Fig. 3a). Second, temporal order memory (i.e., recency discrimination) was significantly impaired for boundary-spanning pairs relative to same-context pairs, $t(33) = -4.77$, $p < 0.001$, $d = 0.82$, [CI: -0.13, -0.052] (Fig. 3b). Additionally, temporal order memory was significantly above chance for same-context pairs ($p < 0.05$) but not for boundary-spanning pairs ($p > 0.05$).

Finally, source memory for the tone/ear paired with each item was significantly higher for items immediately following a boundary compared to those appearing in other event positions, $F(3,25) = 8.68$, $p < 0.001$, $\eta^2 = 0.51$ (Fig. 4). Specifically, source memory was enhanced for boundary items relative to items

appearing in the same context, $p < 0.001$ [CI: 0.032, 0.12], as well as items from the very beginning, $p = 0.021$ [CI: 0.009, 0.15], and end of each list, $p = 0.023$ [CI: 0.007, 0.13].

**Experiment 2: Behavior.** The results from Experiment 1 validated our auditory event boundary manipulation, showing that auditory context shifts elicit temporal and source memory effects consistent with the growing literature on how event memories emerge from continuous experience (for a review, see ref. [10]). In Experiment 2, we combined this behavioral manipulation with eye-tracking to address our main hypothesis that fluctuations in physiological arousal during encoding, as indexed by pupil diameter, relate to the discretization of mnemonic events. All behavioral procedures were identical to those in Experiment 1.

Replicating the results from Experiment 1, participants were again slower to make the indoor/outdoor judgments at encoding for objects that followed a boundary compared to objects that appeared within the same auditory event, $t(34) = 3.09$, $p = 0.004$, $d = 0.52$, [CI: 19.16, 92.55], and at the end of each list, $t(34) = 2.55$, $p = 0.015$, $d = 0.43$, [CI: 12.00, 105.75] (Fig. 2).

All of the memory results also replicated the findings from Experiment 1. Boundary-spanning item pairs were remembered as having appeared farther apart in time than same-context pairs, despite both item pairs appearing the same distance apart at encoding, $t(34) = 3.18$, $p = 0.003$, $d = 0.54$, [CI: 0.06, 0.27] (Fig. 3a). Temporal order memory was again worse for boundary-spanning item pairs compared to items from the same auditory context, $t(34) = -6.45$, $p < 0.001$, $d = 1.09$ [CI: -0.14, -0.071] (Fig. 3b). As in Experiment 1, source memory was significantly better for boundary items compared to items in other event positions, $F(3,32) = 29.07$, $p < 0.001$, $\eta^2 = 0.73$ (Fig. 4). Specifically, source memory was better for boundary items

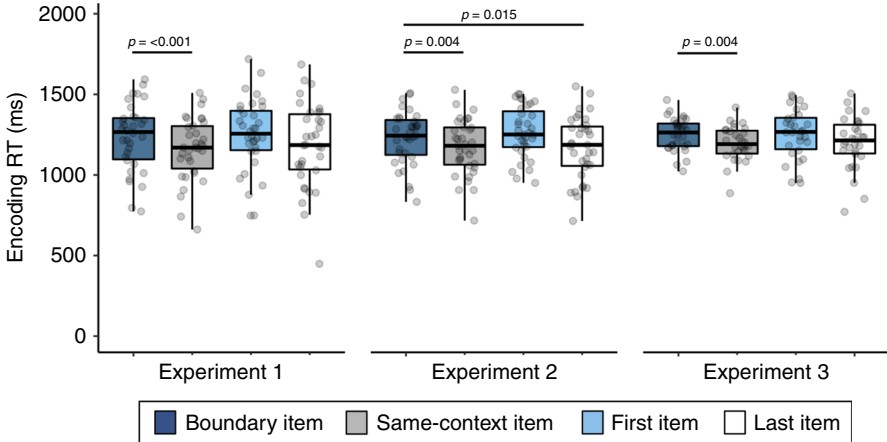

**Fig. 2 Individuals are slower to respond to items appearing just after a tone switch, or event boundary, compared to other items in an event sequence.** Values represent average response times (RTs) for the indoor/outdoor item judgments during sequence encoding. Colored boxplots represent 25th–75th percentiles of the data, the center line the median, and the error bars the s.e.m. Overlaid dots represent individual participants (Experiment 1: $n = 34$; Experiment 2: $n = 35$; Experiment 3: $n = 30$). Two-tailed paired pairwise $t$-tests were performed to test for significant differences between judgment response times for boundary items, first items, last items, and same-context items. These $t$-tests were planned, so no adjustments were made for multiple comparisons. The results for the boundary compared to same-context trial comparisons from the first experiment directly replicated two follow-up experiments. Source data are provided as a Source Data file.

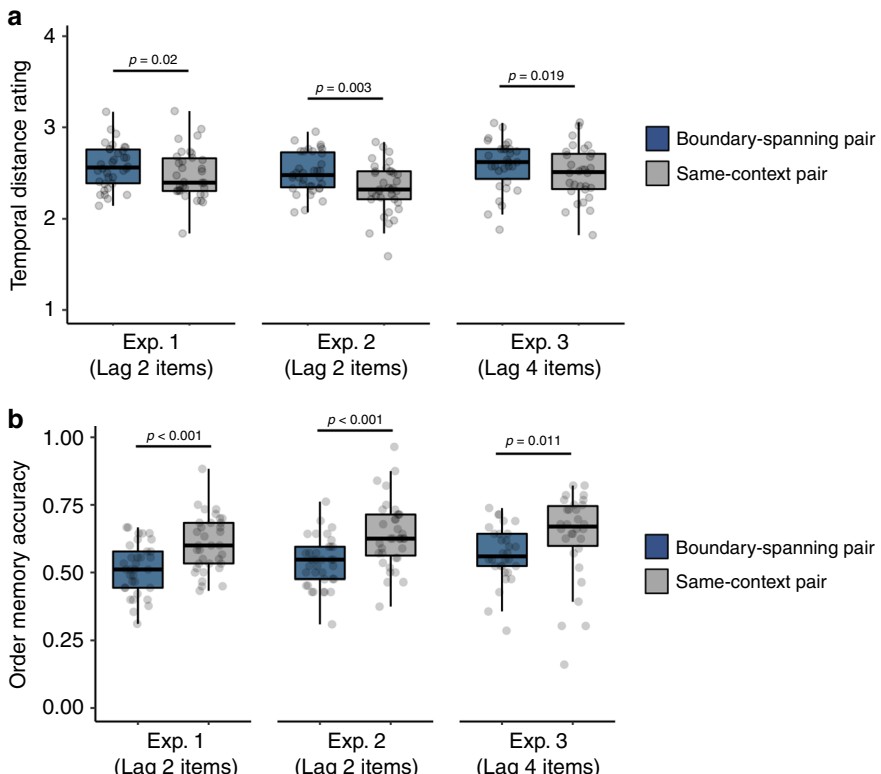

**Fig. 3 Tone switches, or event boundaries, embedded within item sequences lead to impaired order memory and expanded retrospective estimates of temporal distance between item pairs spanning those boundaries. a** Values represent average temporal distance ratings for item pairs from the object sequences. During this temporal memory test, participants rated how far apart the item pairs had appeared in the prior sequence, with choices ranging from 'very close' to 'very far'. The ratings were then converted to a scale ranging from 1 to 4 and averaged together, such that higher values on the $y$-axis reflect more expanded retrospective estimates of temporal distance between item pairs. **b** Values represent average temporal order memory accuracy for item pairs from the object sequences. During this temporal order memory test, participants had to decide which of two items had appeared later (i.e., more recently) in the previous sequence of images. 'Lag' refers to the number of intervening items that had appeared between the to-be-tested item pairs at encoding. For both panels, colored boxplots represent 25th–75th percentiles of the data, the center line—the median, and the error bars—the s.e.m. Overlaid dots represent individual participants (Experiment 1: $n = 34$; Experiment 2: $n = 35$; Experiment 3: $n = 30$). Two-tailed paired $t$-tests were performed to test for significant differences between temporal memory outcomes for boundary pairs compared to same-context pairs. These $t$-tests were planned, so no adjustments were made for multiple comparisons. The results from the first experiment were replicated in two follow-up experiments. Source data are provided as a Source Data file.

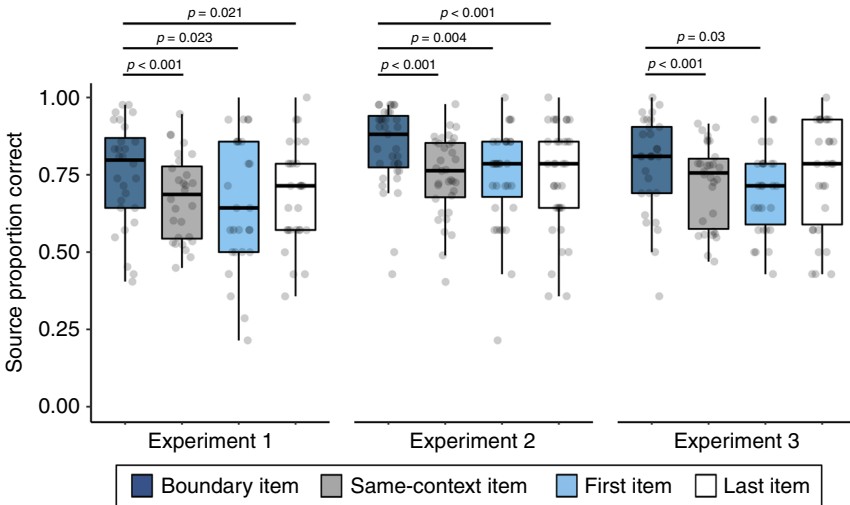

**Fig. 4 Tone switches lead to better source memory for individual items and their accompanying sounds.** During this source memory test, participants had to indicate whether each individually presented item had been paired with a tone played in their left or right ear during encoding. 'Last Item' refers to the last image presented in each 32-item list, whereas 'First Item' refers to the first image presented in each 32-item list. 'Boundary Item' refers to the first object appearing after a tone switch, or event boundary. Colored boxplots represent 25th–75th percentiles of the data, the center line—the median, and the error bars—the s.e.m. Overlaid dots represent individual participants (Experiment 1: $n = 34$; Experiment 2: $n = 35$; Experiment 3: $n = 30$). A repeated-measures ANOVA was performed to test for differences in source memory accuracy by the four different item types. Displayed $p$-values reflect the results of follow-up, two-tailed Bonferroni-corrected pairwise comparisons. The results for the boundary compared to same-context trial comparisons from the first experiment directly replicated two follow-up experiments. Source data are provided as a Source Data file.

relative to items encountered within a stable auditory context, $p < 0.001$ [CI: 0.065, 0.12], as well as items at the very beginning, $p = 0.004$ [CI: 0.021, 0.15], and end of each list, $p < 0.001$ [CI: 0.046, 0.15].

**Experiment 3: Behavior.** The behavioral results of Experiment 2 replicated the behavioral findings in Experiment 1, pointing to the robust effects of event boundaries on the organization of episodic memory. Prior event boundary experiments show that the lag between to-be-tested item pairs at encoding may influence subsequent temporal order memory performance[13,45]. Thus, in the final experiment, we increased the objective distance between the to-be-tested item pairs from two to four intervening items (Supplementary Fig. 1) to examine if (1) the behavioral findings reported thus far are robust to the actual distance between tested item pairs, and (2) potential relationships between arousal at boundaries and temporal memory are diminished.

Replicating the results from Experiments 1 and 2, participants were slower to make indoor/outdoor judgments for boundary items compared to items from a stable context, $t(29) = 3.15$, $p = 0.004$, $d = 0.58$, [CI: 19.14, 90.17] (Fig. 2). Participants again remembered boundary-spanning pairs as having appeared farther apart in time than same-context pairs, $t(29) = 2.48$, $p = 0.019$, $d = 0.45$, [CI: 0.015, 0.16], despite, again, the actual distance being matched (Fig. 3a). Temporal order memory was again worse for boundary-spanning pairs than for same-context pairs, $t(29) = -2.72$, $p = 0.011$, $d = 0.50$, [CI: −0.12, −0.016] (Fig. 3b). Also replicating the results from Experiments 1 and 2, source memory was significantly better for boundary items compared to other item types, $F(3,27) = 8.56$, $p < 0.001$, $\eta^2 = 0.49$ (Fig. 4), with memory being significantly higher for boundary items compared to items at other within-event positions (i.e., same-context items), $p < 0.001$ [CI: 0.027, 0.10] as well as for items appearing at the very beginning of each list, $p = 0.03$ [CI: 0.005, 0.13]. Source memory was not significantly better for boundary items compared to the last items in each list, $p > 0.05$ [CI: −0.044, 0.092].

**Pupil dynamics track event structure.** Across all three experiments, we found reliable evidence that context shifts modulate both temporal and non-temporal features of episodic memories. Next, we tested our main hypotheses that these memory-structuring effects relate to fluctuations in pupil-linked arousal signals across sequence encoding. To this end, we analyzed the pupil data from Experiments 2 and 3 using a temporal PCA to assess whether transient changes in arousal at event boundaries, as indexed by tone-triggered pupil dilation, relate to the effects of boundaries on temporal and source memory.

Across both eye-tracking experiments (Experiments 2 and 3), fluctuations in pupil diameter appeared to be sensitive to event structure during encoding (Fig. 5a, b). That is, although pupil size was dynamically modulated by the occurrence of *all* items and tones during sequence learning, transient spikes in pupil dilation were most robust at boundaries.

To quantify the specific effect of boundaries on pupil size, we compared mean pupil dilation responses to the tone switches (boundary tones) versus pupil responses to repeated tones occurring within a stable auditory context (i.e., tones preceding items 2 through 8 in an event; Fig. 5a). We first established that, across both eye-tracking studies, boundary tones, $t(64) = 11.97$, $p < 0.001$, $d = 1.48$, [CI: 221.56, 310.29], and same-context tones, $t(64) = 4.29$, $p < 0.001$, $d = 0.53$, [CI: 33.25, 91.22], elicited increased pupil dilation compared to baseline. Confirming our hypothesis, we also found that pupil dilation was significantly greater for boundary tones than for same-context tones, $t(64) = 9.75$, $p < 0.001$, $d = 1.21$, [CI: 161.97, 245.40]. All of these results remained significant when analyzing data from Experiments 2 and 3 separately (all $p$'s < 0.05). These results demonstrate that auditory event boundaries were indeed salient and triggered a momentary increase in physiological arousal.

**Boundaries modulate temporal features of pupil dilation.** Next, to further understand how boundaries modulate pupil dilation measures, as well as how this may impact ongoing cognitive processing, we performed a temporal PCA on the preprocessed pupil data. Prior research suggests that there are distinct temporal

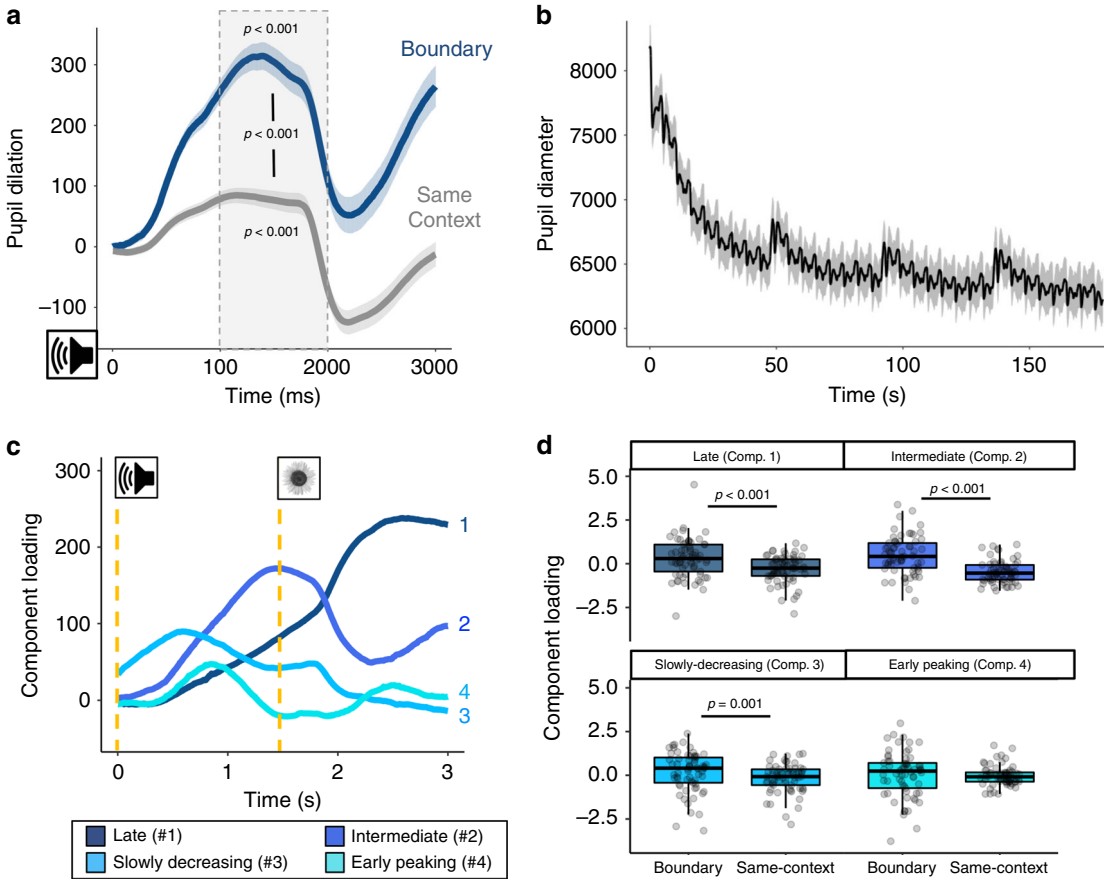

**Fig. 5 Event boundaries modulate temporal characteristics of pupil dilation. a** The effects of boundary (tone switches; blue) versus same-context tones (repeated; gray) on pupil dilation. Shaded areas represent s.e.m. **b** Average fluctuations in pupil diameter across the encoding sequence, averaged across all blocks, experiments, and participants. The gray shaded area represents s.e.m. **c** Temporal features of tone-evoked pupil dilation identified by a temporal principal component analysis (PCA). The PCA revealed four significant features of pupil dilation that had distinct shapes over time. Component loadings reflect "raw" values from the rotated solution, so are on the same scale as the original inputs. Gold dashed lines signify the onsets of the tones and their subsequent images. **d** The boundary tones (blue colors) versus same-context tones (gray) differentially modulated loading scores for the first three pupil components but not the fourth component. Colored boxplots represent 25th–75th percentiles of the data, the center line the median, and the error bars the s.e.m. Overlaid dots represent individual participants (Experiment 2: $n = 35$; Experiment 3: $n = 30$). Two-tailed paired pairwise $t$-tests were performed to test for significant differences between factor-loading values for boundary trials compared to same-context trials. No adjustments were made for multiple comparisons. Source data are provided as a Source Data file.

characteristics of pupil dilation that are regulated by different cognitive and neurophysiological processes, including cognitive control, motor responses, and salience detection[39–44]. For instance, oddball stimuli have been shown to modulate an early-peaking (~800 ms) component under moderate light, but not under darker conditions when parasympathetic tone is minimal[42,43]. This finding suggests that this early-peaking aspect of pupil dilation may specifically index parasympathetic inhibition, which elicits pupil dilation via relaxation of the sphincter muscle[46]. Based on this work, we leveraged PCA decomposition to identify different temporal characteristics of stimulus-triggered pupil dilation, and to link ostensible cognitive components of these dissociable physiological responses to different episodic memory outcomes.

To identify different sub-components of pupil dilation, we averaged all of the pupil samples across the time-window of the tone-evoked pupil dilations (i.e., onset of tone plus three seconds; see Fig. 5a) across participants and experiments. Importantly, the PCA was completely data-driven and agnostic to condition (i.e., boundary versus same-context tones). The PCA revealed four principal components that accounted for significant variance in tone-evoked pupil dilations (Fig. 5c). The temporal features of

these components, including their latencies-to-peak and the amount of variance they accounted for, were as follows: (1) a late component (2424 ms; 76.09% variance); (2) an intermediate component (1316 ms; 16.35% variance); (3) a slowly decreasing component (308 ms; 3.01% variance); and (4) an early peaking component (800 ms; 2.66% variance). These pupil components were highly consistent with prior work applying PCA to pupil data, including a biphasic response that may signify separable contributions of the parasympathetic and sympathetic nervous systems to pupil size[41–44].

We next asked whether the degree to which individual participants loaded onto these pupil components was modulated by event boundaries. Here, loading refers to a measure of how much participants exhibited these distinct temporal patterns of pupil dilation in response to tones. Because we knew which data-points belonged to each condition, we were then able to compare loading differences between boundary and same-context tones. As shown in Fig. 5d, boundaries significantly modulated loading values for the late component, $t(34) = 4.81$, $p < 0.001$, $d = 0.81$ [CI: 0.31, 0.75], intermediate component, $t(34) = 6.00$, $p < 0.001$, $d = 1.01$ [CI: 0.67, 1.35], and slowly decreasing component, $t(34) = 3.77$, $p = 0.001$, $d = 0.64$ [CI: 0.20, 0.68] across the two

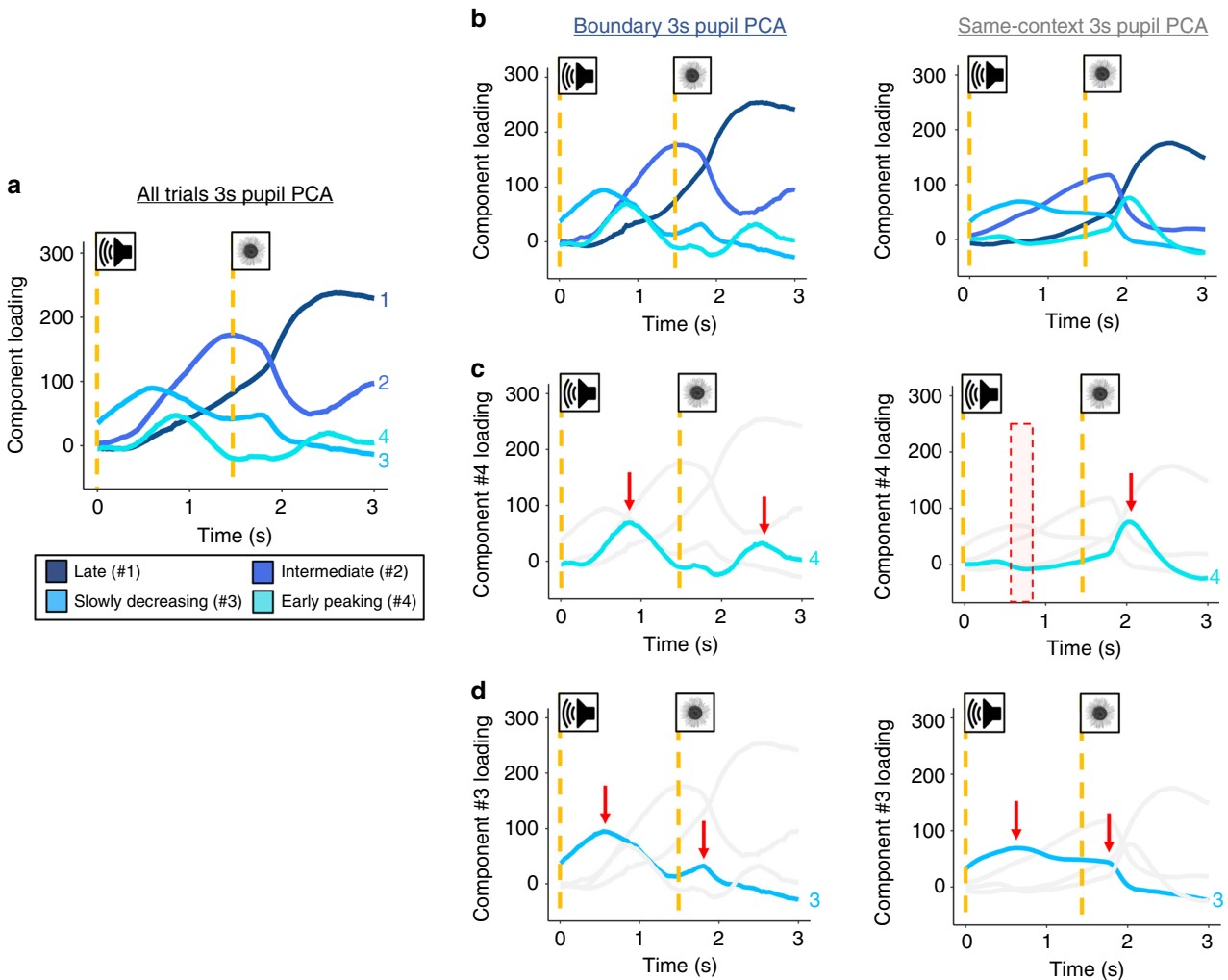

**Fig. 6 Temporal characteristics of pupil dilation evoked by tones and their subsequent images reflect different motor and anticipatory aspects of the sequence learning task. a** A temporal principal component analysis (PCA) on the pupil dilation data revealed four pupil components that had distinct shapes across time. **b** Two follow-up PCAs separated by condition help illustrate how loading on these components differed between event boundary trials and same-context trials. Vertical dashed lines signify the onsets of the tones and their subsequent images. In both conditions, most of the temporal characteristics of the pupil component loadings were qualitatively similar, except for the early-peaking component (component #4; turquoise). **c** To better illustrate these differences, only component four's loading time-course is displayed. The plot reveals evidence of this early peaking component in response to the boundary tones, boundary images, and same-context images (red arrows). However, this component did not peak in response to same-context tones (red shaded area), which was the only stimulus type (tone or image) that did not require a motor response. **d** Only the slowly decreasing component (#3; sky blue) is highlighted to illustrate potential differences in its loading patterns over time. A peak in this component's loadings is identifiable for each tone and image type (red arrows).

experiments. By contrast, there was no significant effect of boundaries on the early-peaking component loadings, $t(34) = 0.29$, $p = 0.77$, $d = 0.049$ [CI: $-0.43$, $0.57$]. The same results were also obtained when we examined the loadings from Experiments 2 and 3, separately (components 1, 2, and 4: $p$'s $< 0.05$; component 3: $p$'s $> 0.05$). Thus, boundaries significantly enhanced some but not all components of pupil dilation, suggesting that context shifts may engage specific mental processes and autonomic pathways.

**Temporal features of pupil dilation reflect task relevance.** To shed additional light on the functional significance of the pupil components, we also examined qualitative differences in the pupil response to auditory tones and their subsequent images in boundary and same-context trials, separately (Fig. 6a, b). Increasing evidence suggests that context shifts modulate memory for and attention to information presented at

boundaries[16,18,47]. Because boundary information may be processed differently in attention and memory than same-context information, we reasoned that the "boundary-ness" of both the context shift and the following item may relate to a spike in arousal (captured by pupil dilations to both the tone and its following item). We expected that the response properties of these components may differ according to whether or not the task/context information required a corresponding shift in behavior, such as a motor response.

Across the two conditions (same-context and boundary trials), there are four timepoints during which a stimulus is presented (two tones and two images), and all but one of those timepoints also requires a motor response. Namely, only the same-context tone does not require any re-mapping of motor responses or actions (i.e., stimulus judgment button press or switching of hands). Thus, if a particular component is related to motor responses, we did expect to see the least evidence of this

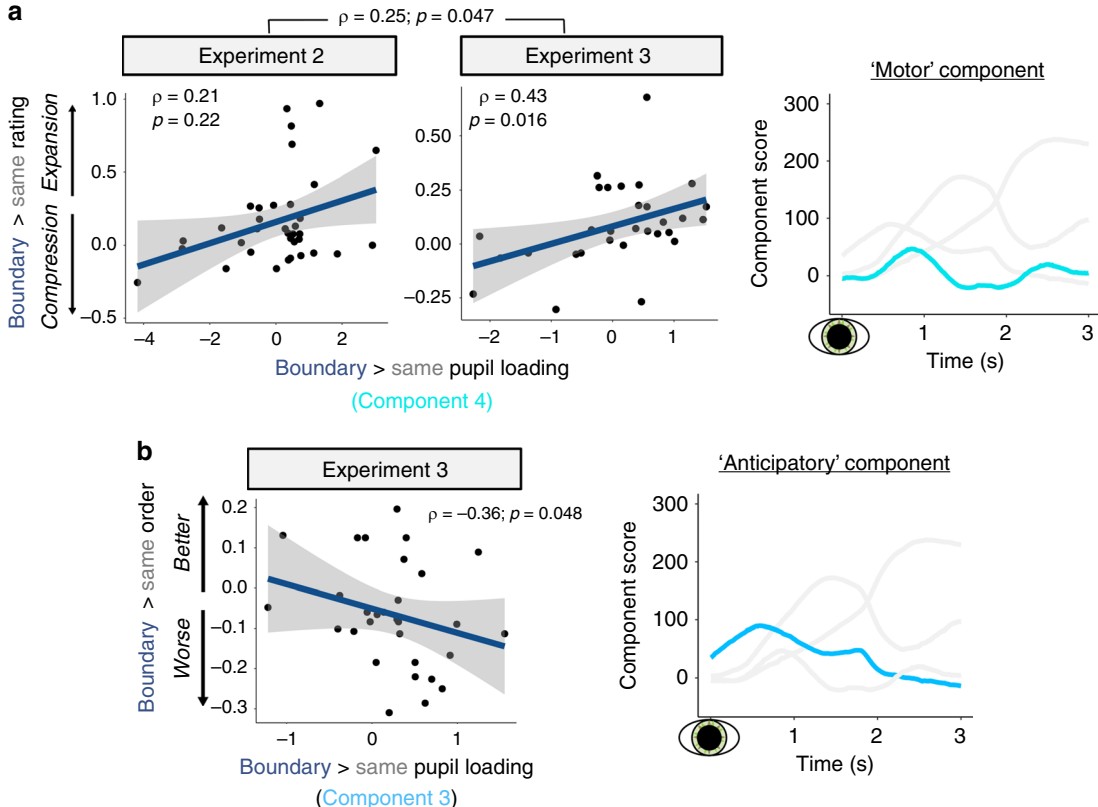

**Fig. 7 Different temporal characteristics of boundary-triggered pupil dilation relate to subjective and objective aspects of temporal memory. a** Association between boundary-modulated loadings on the early-peaking pupil component (#4; turquoise) and boundary-related effects on retrospective estimates of temporal distance between item pairs. Higher values on the *y*-axis reflect more expanded estimates of average temporal distance for boundary-spanning pairs compared to same-context pairs. This pupil–memory relationship was significant across both eye-tracking studies (Experiments 2 and 3) and significant in Experiment 3 alone. **b** Association between boundary-modulated pupil loadings on the slowly decreasing pupil component (#3; sky blue) and boundary-related effects on temporal order memory between item pairs in Experiment 3. Lower values on the *y*-axis reflect worse average temporal order memory for boundary-spanning item pairs compared to same-context pairs. Gray shading in all three regression plots represents 95% confidence interval. Overlaid black dots represent individual participants (Experiment 2: *n* = 35; Experiment 3: *n* = 30). *p*-values were derived from two-tailed tests, and no corrections were made for multiple comparisons. Source data are provided as a Source Data file.

component during same-context tones when no action is required. Consistent with this possibility, the same-context tone was the only stimulus timepoint where this early peaking pupil component did not show a peak (see shaded red window; Fig. 6c). By contrast, the slowly decreasing component (component #3, sky blue; Fig. 6d) was evident during all stimulus onsets.

To quantify boundary-related effects on loadings during the tone period, we next limited the pupil-sampling window of the PCA analyses to 1.5 s (thereby excluding any pupil effects driven by the images). The majority of these confirmatory analyses are reported in Supplementary Methods. Briefly, the 1.5-s PCA replicated three of the four significant pupil components from the 3-s PCA—including, importantly, this early peaking component. Two-tailed paired *t*-tests revealed that boundary tones significantly increased loading on this component relative to same-context tones, $t(34) = 6.00$, $p < 0.001$, $d = 1.01$, [CI: 0.67, 1.35]. This result lends additional support to the idea that this pupil component relates to the motor re-mapping that occurs selectively at boundary tones and not at same-context tones (see Supplementary Fig. 2b for more detail).

**Spikes in arousal index mechanisms of memory separation.** To test our key hypothesis that spikes in arousal are associated with event memory separation, we performed Spearman's rank

correlation analyses between loadings on the four pupil components and the three episodic memory outcomes: temporal distance ratings, temporal order memory, and source memory.

Across both eye-tracking experiments, there was a significant positive association between pupil dilation at boundaries and temporal distance memory. Specifically, as shown in Fig. 7a, individuals who exhibited a time expansion effect in memory for boundary-spanning pairs also showed more evidence of the early-peaking pupil component in response to boundaries versus same-context items ($\rho = 0.25$, $p = 0.047$). The relationship between the early peaking pupil component and temporal distance memory was significant in Experiment 3 ($\rho = 0.43$; $p = 0.016$) and showed a trend towards significance in Experiment 2 ($\rho = 0.21$; $p = 0.22$) when these experiments were analyzed separately.

Next, we examined the relationship between the pupil dilation components and temporal order memory performance. We did not observe any significant pupil–memory associations when the data were collapsed across the two eye-tracking studies or in Experiment 2 alone. However, in Experiment 3, where the objective distance between the test pairs was the largest (i.e., four intervening items in Experiment 3 versus two in Experiment 2), we identified one significant relationship. As shown in Fig. 7b, individuals who exhibited worse temporal order memory for boundary-spanning item pairs compared to same-context pairs also showed more evidence of the slowly decreasing pupil

component ($\rho = -0.36$, $p = 0.048$) at boundaries relative to non-boundaries. Additionally, there were no significant associations between any of the pupil components and enhanced boundary-item source memory (all $p$'s > 0.05).

In summary, our findings suggest that potentially unique cognitive and pupil-linked processes, as indexed by distinct temporal characteristics of pupil dilation, may influence subjective and objective temporal aspects of episodic memory. These different pupil–memory relationships support our key hypothesis that dynamic fluctuations in pupil size are sensitive to cognitive and arousal-related processes triggered by context shifts, which in turn elicit more separated, or segmented, memories of temporally adjacent experiences.

On one hand, the influence of boundaries on subjective temporal memory was related to an early peaking pupil component that is engaged during a time window consistent with motor responses (~600–900 ms[41,48]). Indeed, we only found evidence of this pupil response at timepoints that required a motor response (hand change or button press). On the other hand, the influence of boundaries on temporal order memory was related to a slowly decreasing pupil dilation component that is triggered just prior to the onset of an impending boundary and its first memorandum (e.g., ref. 40). Interestingly, we did not observe any relationships between loading on the pupil components and enhanced source memory at boundaries, suggesting that boundary-elicited arousal responses may specifically relate to processes that facilitate the temporal organization of events in memory.

**Average phasic and tonic arousal do not account for memory.** Next, in a series of three exploratory analyses, we examined how trial-level fluctuations in phasic and tonic states of arousal, indexed by pupil dilation and overall pupil diameter, respectively, relate to subsequent memory. The PCA dissociated how different temporal characteristics of boundary-evoked pupil dilation relate to memory across participants. However, it is also possible that trial-level changes in tonic (global) and/or phasic (transient, stimulus-evoked) arousal states modulate event segmentation in memory. This idea is inspired by evidence that tonic arousal states also influence memory and decision processes, perhaps even in different ways than stimulus-evoked responses[49]. To explore these possibilities, we used linear-mixed modeling to see if changes in phasic (indexed by tone-evoked pupil dilation) and tonic arousal (indexed by average pupil size during the pre-tone baseline period) were associated with subsequent memory outcomes (Supplementary Analysis 2).

Briefly, the mixed modeling results revealed no significant relationships between either of these trial-level measures of pupil size and memory (Supplementary Fig. 3). This suggests that a trial-level pupil-averaging approach may obscure more nuanced, arousal-linked mechanisms detected by the PCA. This underscores the importance of dissociating the roles of different arousal and cognitive processes in episodic memory segmentation, which can be accomplished using PCA decomposition.

**Boundary salience does not relate to pupil size or memory.** Prior work suggests that the magnitude of prediction errors, which have been theorized to induce event boundaries, scales with event segmentation effects in memory[18]. Moreover, prediction errors are associated with activation of brainstem arousal systems. Thus, we leveraged the varying magnitude in tone pitch changes across boundaries (e.g., ranging from 100 to 500 Hz) to see whether the amount of change at a context shift impacts later memory separation (Supplementary Methods).

A trial-level linear-mixed modeling analysis revealed no significant effects of tone pitch changes at boundaries on any memory outcomes. These tone changes were also not correlated with trial-level changes in pupil dilation, suggesting that the degree of context change did not trigger different levels of arousal (Supplementary Fig. 4b). There also was no significant relationship between tone pitch changes at boundaries and memory outcomes for items at those boundaries. Taken together, these findings suggest that the change in the sensory properties of the tone itself did not elicit different levels of phasic arousal or modulate episodic memory (Supplementary Fig. 4b). Rather, the task relevance of the tone switches appeared to be sufficient to elicit memory separation. This null result, however, does not exclude the possibility that the range of pitches (500–1000 Hz) in these experiments was too limited to yield enough variability to account for discrete changes in later memory.

**Temporal stability of arousal indexes memory integration.** In the final exploratory pupil analysis, we tested if fluctuations in arousal over more prolonged periods of time support temporal memory integration (Supplementary Methods). Behavioral evidence suggests that dynamic fluctuations in external contexts (e.g., space) over time modulate cognitive processes that link together sequences of information[10]. Here we reasoned that if arousal itself is considered an internal contextual state, its stability across time may be related to the integration of sequential representations in memory. Indeed, we found that lower trial-by-trial variability in pupil diameter between the to-be-tested item pairs at encoding (indexed by standard deviation in pupil size) was related to more compressed retrospective estimates of temporal distance and better temporal order memory (Supplementary Fig. 5).

## Discussion

Addressing how the brain transforms continuous experience into memorable episodes is foundational to our understanding of learning and memory. Here, we provide evidence that (1) dynamic fluctuations in pupil size are sensitive to the structure of unfolding experiences, and (2) these pupil-linked autonomic arousal changes are in turn related to how those experiences become encoded and represented as discrete memories. We first demonstrate that salient context shifts during sequence learning elicit increased pupil dilation. Next, we use PCA decomposition to show that dissociable temporal components of this pupil response—including those previously linked to increased cognitive control, motor responses, and response anticipation—are associated with temporal memory outcomes that are consistent with the discretization of events in long-term memory. These findings underscore that pupil-linked arousal mechanisms engaged during ongoing experience are sensitive and/or contribute to the segmentation of events in memory.

A growing literature suggests that temporal context stability (e.g., remaining in the same room for an extended period of time) may provide scaffolding for linking sequential experiences together in memory (for a review, see ref. 10). Conversely, a context shift may drive memories of adjacent experiences farther apart. Our findings lend support to this idea by showing that auditory event boundaries elicited greater subjective separation in memory between temporally adjacent events, impaired temporal order memory, and enhanced source memory for local contextual features. These data expand upon earlier work using visual category or perceptual shifts as event boundaries by demonstrating that stability and change in auditory contexts elicit similar memory integration and separation effects[1,4,5,9–18]. Many types of context shifts can thereby drive event separation in memory, implicating

a shared underlying process that is not limited to one sensory modality.

Interestingly, we also demonstrate that the stability of pupil-linked arousal states over time helps integrate dynamic experiences into memorable events. Previous work suggests that there are many organizational principles by which temporally-extended experiences cluster together in memory, including temporal context[50], semantics[51], and overlapping perceptual features[1,13,16]. However, this body of work mostly considers contextual information that is either theoretical (e.g., mental context[50,52]) or comes from the external environment (e.g., categories, colors, space[9]). Our results suggest that, like other forms of context, arousal might be a strong feature of an internal contextual state that binds information together in memory.

Using PCA, we found that momentary increases in different temporal characteristics of pupil dilation at boundaries were associated with subjective and objective markers of event memory separation. The PCA first revealed prototypical temporal features of pupil dilation, which have been previously associated with different autonomic pathways that regulate pupil size[41,42]. These features of pupil dilation have also been linked to different aspects of cognition, such as sustained cognitive processing, response anticipation, and motor responses[39–44], suggesting that multiple arousal-related mental processes are engaged by boundaries in ways that relate to later memory segmentation.

Across two eye-tracking experiments, we found that an early peaking component of pupil dilation, which is thought to reflect parasympathetic regulation of pupil size under conditions that require sustained cognitive processing or a motor response[41,42], was related to greater retrospective estimates of temporal distance. Consistent with the idea that this component reflects a decision and/or motor process, this component was only absent during the same-context, or repeated, tones during which time no hand-switch or response was required. Importantly, the extent to which individuals engaged this component at boundaries was also associated with subjective distortions in memory; namely, individuals who showed greater engagement of this pupil response at boundaries were also more likely to remember recent events as having occurred farther apart in time. According to a seminal model of event segmentation, people are constantly maintaining an active representation of the world, or event model, which requires sustained cognitive processing[7]. At an event boundary, even greater cognitive control or sustained processing may be necessary to update this event model with new contextual information. In the current task, the remapping of motor responses at boundaries provides essential, task-relevant contextual information that helps to distinguish one event from the next. One possibility, then, is that this early peaking pupil response may be capturing the intensity of this event model-updating process during initial experience, which in turn predicts how separated adjacent representations become in memory space.

In contrast to the temporal distance ratings, we found that boundary-related impairments in an objective memory measure, temporal order memory, were associated with engagement of a different pupil component that peaked around the time of tone onset and slowly decreased thereafter. We interpret this slowly decreasing pupil dilation component (#3 in the current study) as signifying anticipatory arousal, given evidence that it is triggered in preparation of impending responses[40]. When we accounted for the combined pupil response to both the onsets of the tones and their subsequent images (i.e., the 3 s window PCA), we found more evidence of this slowly decreasing component during event boundaries compared to same-context trials. This result suggests that the initial information presented after a boundary may also be processed differently than same-context information, perhaps

because it is the first piece of information that becomes bound to a novel context (i.e., it may define or become an important part of a new event model[7]). In our experiments, all events were of the same length, and there was an equal amount of time between each item in the sequences. The anticipatory pupil response may thereby reflect participants' ability to predict when an impending tone switch and image would occur. From this perspective, predictable changes in the environment could be used to proactively allocate ongoing experiences into contextually appropriate memory representations. Indeed, individuals can still perceive discrete events even when the transition between them is predictable[53]. This pupil–memory relationship was only observed in the experiment with a larger number of intervening items between the to-be-tested item pairs (Experiment 3), suggesting that this mechanism may have a more significant ripple effect on information encountered farther away in time from the boundary.

Importantly, some of our interpretations concerning the functional significance of these pupil components are drawn from previous studies using PCA. To further explore if anticipating a boundary during ongoing experience can guide event segmentation in memory, future studies could test whether this pupil–memory association disappears when context shifts are more or less predictable by varying the number of items and/or the temporal distance between them within a sequence. One interesting possibility is that unpredictable event boundaries also elicit event segmentation in memory by triggering arousal-related prediction error signals that facilitate event-model updating (e.g., ref. [54]). However, this model-updating process may be signaled by other temporal components of pupil dilation.

Research in both animals and humans suggests that variations in pupil size may be a reliable biomarker of locus coeruleus-norepinephrine (LC-NE) system activity[55,56], which is known to modulate attention and the encoding of salient events[28,29]. LC neurons regulate pupil diameter both directly and indirectly through sympathetic and parasympathetic nervous system pathways, respectively[57,58]. Thus, the functional neuroanatomy of the noradrenergic system makes it well positioned to support the observed boundary-related effects on temporal memory. Interestingly, core models of noradrenergic function propose that NE release helps to orchestrate a "network reset" that reorients attention and functional brain networks to process salient environmental changes[27]. This model bears a striking resemblance to a theoretical reset signal that is thought to trigger event segmentation processes at boundaries[7,54]. Furthermore, it is well established that under arousal, NE modulates memory processes in the hippocampus[59], a brain region that is integral to binding contextual and temporal information in memory[60–62]. Collectively, our results raise the possibility that arousal-related LC activity not only signals event boundaries, but also modulates processes that shape the temporal structure of memory downstream.

Of course, another possibility is that our results signify the contributions of multiple neuromodulatory systems to episodic memory organization. For instance, acetylcholine (ACh) release also co-occurs with pupil dilation[56] and regulates pupil size exclusively via parasympathetic nervous system pathways[57]. Given that we found a strong association between a parasympathetic-related feature of pupil dilation (the early-peaking component) and temporal distance ratings, one possibility is that ACh may specifically modulate subjective representations of time. Like NE, ACh also influences hippocampal-encoding processes, particularly those that support the separation of experience into distinct memories[63]. Future work could investigate these neuromodulatory mechanisms using a combination of fMRI, eye-tracking, and pharmacology.

The present findings have important implications for improving learning in both educational settings and everyday life. For instance, the ability to perceive and segment events is associated with enhanced memory for those events, even up to one month later[64–66]. The boundary-related impairments in temporal order memory we report might thereby be beneficial for structuring event memories in ways that improve long-term recall. This memory-structuring process seems to be driven, in part, by individuals anticipating and responding to meaningful contextual changes in the environment, as signaled by the activation of specific sub-components of pupil dilation. Our work highlights that arousal fluctuations play an important role in signaling and/ or supporting this organizational process, and suggests that manipulating the structure of learning with arousing context shifts may be especially effective for enhancing long-term memory in the classroom and beyond.

## Methods

**Experiment 1: Participants**. Thirty-four individuals (23 women; $M_{age} = 23.26$, $SD_{age} = 4.52$) were recruited from the New York University Psychology Subject Pool and nearby community to participate in this experiment. All participants provided written informed consent approved by the New York University Institutional Review Board and received monetary compensation for their participation. A power analysis was performed on data from a similar event boundary experiment to estimate the appropriate sample size[13]. With an $\alpha = 0.05$ and power = 0.80, we needed 28 participants to obtain a large effect size ($d = 0.80$; Cohen's criteria[67]) for the temporal order memory effect (G*Power 3.1).

Additional participants were recruited in case of poor memory performance, potential withdrawal from experiment, or an inability to perform the task. We also expected overall temporal memory performance to be worse in the current experiment compared to the results reported in Dubrow and Davachi[13], given that the sequence lists had eight additional items and there was a shorter lag between to-be-tested item pairs (two vs. three items). All eligible individuals had normal or normal-to-corrected vision and hearing, and were not taking beta-blockers or psychoactive drugs. For the source memory analysis, six participants were excluded from data analysis due to a programming error.

**Materials**. The object stimuli consisted of 480 images of everyday objects on a white background. These images were selected from previous datasets[68,69]. Each image was resized to be $300 \times 300$ pixels and rendered in grayscale. To control for non-cognitive-related effects on pupil size, the luminance of all object images and fixation screens was normalized using the SHINE toolbox in MATLAB. To manipulate the surrounding context during visual sequence learning, six 1 s pure tones with sine waveforms of different frequencies (500, 600, 700, 800, 900, 1000 Hz) were generated using Audacity (https://www.audacityteam.org/). These frequencies were chosen such that sounds were discriminable from one another and were salient enough to maintain participants' attention.

**Procedure**. In the current study, we performed three separate behavioral experiments in which we queried different aspects of episodic memory from a sequence-learning task. Building on prior studies from our lab, we developed a paradigm in which event boundaries within an image sequence were defined as a switch from one stable auditory context—in which the same tone was played in the same ear—to another (Fig. 1). The experiments were conducted using E-Prime Version 2.0.

**Sequence encoding**. For each sequence, participants viewed a series of 32 grayscale, luminance-normed images of objects. Each image was presented in the center of a gray background for 2.5 s. During the inter-stimulus interval (ISI) between each object, a black fixation cross was displayed in the middle of the screen for 3 s. Half-way through each ISI, or 1.5 s post-image, a 1-s pure tone was played in either the participant's left ear or right ear. The ear that the tone played is indicated to participants which hand they should use to make their indoor/outdoor judgment (e.g., left ear = left hand). Specifically, the participant had to indicate via button press whether the displayed object would more likely be found in an indoor or outdoor setting.

Importantly, the specific tone/ear pairing heard before each object remained the same for eight successive object images, which served to create a stable auditory context, or event. After the eighth item in each event, the tone switched to the other ear and changed in pitch, creating an event boundary. This new tone/ear pairing then remained the same for the next eight items, and so on and so forth. There were three-event boundaries per list, creating a total of four auditory sub-events per list. The tone frequencies were pseudorandomized across lists, such that no tones of a given frequency were presented in more than one of the four sub-events within a list (i.e., in a given list, tones that were 700 Hz were not heard in more than one eight-item event). The ear that the tones first played in was also counterbalanced

across lists. Each participant viewed a total of 15 lists/sequences. The first list served as a practice block, allowing participants to become accustomed to the encoding and memory tasks. The data from the remaining 14 lists were included in all subsequent analyses.

**Delay distractor task**. To create a 45-s study-test delay, and to reduce potential recency effects, participants performed an arrow detection task after each sequence. In this phase, a rapid stream of either left-facing (<) or right-facing (>) arrow symbols appeared in the middle of the screen for 0.5 s each. These arrow screens were separated by 0.5-s ISI screens with a centrally presented black fixation cross. Participants simply had to indicate which direction the arrow was pointing via button press as quickly as possible.

**Temporal memory tests**. Following the distractor task, we tested three aspects of episodic memory. These tests were also divided into two different blocks of trials. The first block included two temporal memory tests, and the second block included source memory judgments. In the temporal memory block, participants were shown pairs of items from the prior sequence. First, we queried temporal order memory by having participants indicate which of two probe items from the prior sequence had appeared more recently (Fig. 1). After this choice, the same pair of items remained on-screen, and participants had to rate the temporal distance between the two items. For this temporal distance rating, participants could rate item pairs as having appeared very close, close, far or very far apart in the prior sequence (e.g., ref. [1]). Crucially, each pair of items had always been presented with two intervening items during encoding, and were thereby always encountered the same objective distance apart. Thus, any differences in temporal distance ratings between the two pair types were completely subjective. There was no time limit for each response. To test our hypothesis that event boundaries alter temporal memory organization, we considered two types of item pairs: (1) items that had appeared within the same auditory event (same-context pairs; four trials per list) and (2) items that had spanned an intervening tone switch (boundary-spanning pair; three trials per list).

**Source memory test**. In the second block of trials, we queried source memory for all of the items that had not appeared in the temporal memory test block (18 items total; Supplementary Fig. 1). Each item was displayed individually in the center of a gray screen. Participants had to indicate whether each object had been paired with a tone played in their left ear or in their right ear. There was no time limit for each choice, and the order of the items was re-randomized so that it did not match the presentation order from encoding. Examples of the item positions that were tested after each sequence are displayed in Supplementary Fig. 1 (white boxes).

**Memory analyses**. Temporal order memory performance was calculated as the proportion of correct recency discriminations within each condition. To examine the effects of boundaries on order memory, these values were then submitted to a repeated-measures analysis of variance (rm-ANOVA; context condition: boundary-spanning pair, same-context pair). For temporal distance memory, the four possible ratings were converted to a scale ranging from 1 (very close) to 4 (very far) and then averaged together. The resulting distance values were submitted to a rm-ANOVA (context condition: boundary-spanning pair, same-context pair). Effect sizes for these tests are reported as Cohen's $d$.

Source memory performance was calculated as the proportion of correctly remembered ear/sound side within each condition; specifically, whether participants successfully remembered whether an individual item had been paired with a tone played in the left ear or right ear. The source memory data were also analyzed using a rm-ANOVA, except this time the first item and the last item from each list were separated from the same-context condition to mitigate any potential recency or primacy effects on source memory (context condition: boundary item, same-context item, first item, last item). Bonferroni-correct, planned follow-up two-tailed paired $t$-tests were then used to compare source memory accuracy for boundary items versus same-context items, as well for the first and last items in each list. All statistical analyses were performed using SPSS Version 25.

**Experiment 2: Participants**. Forty individuals were recruited from the New York University Psychology Subject Pool and nearby community to participate in this experiment. All participants provided written informed consent approved by the New York University Institutional Review Board and received monetary compensation for their participation. Inclusion criteria were the same as in Experiment 1. Five participants were excluded from data analysis: three participants withdrew mid-way through the experiment, one participant failed to follow task instructions, and the eye-tracker malfunctioned for one participant. Thus, data from 35 individuals were analyzed in this experiment (24 women; $M_{age} = 22.57$, $SD_{age} = 4.24$).

**Experiment 2: Procedure and memory analyses**. The task for Experiment 2 used the same procedure as Experiment 1, with the addition of eye-tracking.

**Experiment 3: Participants**. Thirty-eight individuals were recruited from the New York University Psychology Subject Pool and nearby community to participate in this

experiment. All participants provided written informed consent approved by the New York University Institutional Review Board and received monetary compensation for their participation. Inclusion criteria were the same as the first two experiments. A total of eight participants were excluded from data analysis: Four participants had poor eye-tracking quality (fewer than 50% valid samples) and four participants withdrew midway through the experiment. Thus, data from 30 individuals were analyzed in this experiment (21 women; $M_{age} = 23.87$, $SD_{age} = 5.55$).

**Experiment 3: Procedure and memory analyses**. The task for Experiment 3 used the same procedure as Experiments 1 and 2, except for one modification. In this version, we presented four intervening items rather than two intervening items between the item pairs that were subsequently tested during the temporal memory tests (see Supplementary Fig. 1). All of the behavioral analyses were the same as in Experiments 1 and 2.

**Experiments 2 and 3: Eye-tracking methods**. During sequence learning, participants were seated 55 cm from the computer and pupil size was measured continuously at 250 Hz using an infrared EyeLink 1000 eye-tracker system (SR Research, Ontario, Canada). By presenting tones during fixation screens appearing between the objects, we were able to acquire clean measures of pupil dilation that were unconfounded by stimulus brightness and visual complexity. The pupil data were preprocessed and analyzed using in-house code implemented in Matlab 9.4 (MathWorks, Natick, MA). Experiment blocks (sequences/lists) with <50% valid pupil data were removed from the final analysis. Eye-blinks and other artifacts, such as signal dropout, were removed using linear interpolation.

**Pupil dilation analysis**. Tone-evoked pupil dilations were compared for two events of interest: the boundary tone (i.e., tone switch after the eighth item in an event) and the same-context, or repeated, tones (i.e., the seven subsequent tones that remained stable within an event). To measure pupil dilation responses, average pupil diameter was measured from 1 to 2 s after cue onset when the tone-evoked pupil response was most apparent (Fig. 5a). The average pupil size during this time-window was then baseline-normed by subtracting the average pupil size during the 500 ms window prior to tone onset. To examine the influence of event boundaries on pupil dilation, we performed two-tailed paired t-tests comparing average pupil dilation responses between the boundary (i.e., tone switched) and same-context (i.e., tone repeated) trials.

**Pupil dilation temporal PCA**. To analyze how event boundaries altered different components of pupil dilation, we performed a temporal PCA by adapting methods described in Johansson et al.[40]. Experiments 2 and 3 were conducted in the same behavioral testing room and under the same dimly-lit lighting conditions. Thus, we combined the data from the two eye-tracking experiments to take advantage of additional statistical power to detect different pupil components. This also enabled us to draw comparisons between the same components across the two eye-tracking studies. The PCA was performed using SPSS Version 25.

Average baseline-normed pupil dilation was computed for each of the tone types (boundary and same-context trials). These values were then averaged across participants to reduce the amount of input variables (130 variables; 65 participants with two conditions each) and to control for noisy, spontaneous pupil changes on individual trials. All pupil samples across a 3-s window covering the average time-course of tone-evoked pupil dilation (750 pupil samples in total; see Fig. 5a) served as dependent measures in the PCA. This pupil-sampling time window was selected in order to capture the "boundary-ness" of both the tone switch and the first boundary item (i.e., item in position 1 within an 8-item event).

A Varimax rotation was performed on the components output by the temporal PCA, which were defined based on an eigenvalue equal to or greater than the average value of the original variables. An unrestricted PCA using the covariance matrix with Kaiser normalization and Varimax rotation was used on all components to generate maximal component loadings on one component with minimal overlap with other components. The resulting loadings reflect the correlated, temporally dynamic patterns of pupil dilation elicited by the auditory tones. Factor loadings with eigenvalues >1 were analyzed in subsequent analyses (Kaiser criterion[70]).

Because the PCA was data-driven and agnostic to condition, we were able to then examine the relative contributions of loading patterns for each pupil component to same-context versus boundary tones. Two-tailed paired t-tests were performed on these loading values to determine how boundaries modulated different temporal characteristics of pupil dilation. In addition, we performed two follow-up PCA's on the pupil data for the boundary and same-context trials, separately. This was done using the same approach as the original analysis with the data collapsed across both conditions.

**Relationship between pupil components and temporal memory**. To test our key hypothesis that increased arousal at boundaries modulates episodic memory organization, we next performed three multiple linear regression analyses between boundary-related effects on pupil loadings and boundary-related effects on each of the three memory outcomes (temporal distance, temporal order, and source memory) in Experiments 2 and 3. To mitigate the influence of any outlier data-

points on these correlations, we performed Spearman's rank order correlations (two-tailed) between pupil and memory subtraction scores. These values were computed as the boundary minus same-context trial values for both the pupil loadings on each component and performance for each memory measure. These Spearman correlations were first performed by collapsing the data across both experiments. In order to see whether the effects replicated in the two eye-tracking experiments individually, we then analyzed the data from each experiment separately.

**Reporting summary**. Further information on research design is available in the Nature Research Reporting Summary linked to this article.

## Data availability
All behavioral and eye-tracking data are publicly available on the Open Science Framework website '4QZNX'. The source data underlying Figs. 2–4, 5a–d, 6c, d, 7a, b, Supplementary Figs. 2a, b, and 3–5 are provided as a Source Data file. A reporting summary for this article is available as a Supplementary Information file. Source data are provided with this paper.

## Code availability
Codes and scripts will be provided upon reasonable request to the corresponding author. Source data are provided with this paper.

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

## Acknowledgements

This project was funded by federal NIH grant R01 MH074692 to L.D. and by fellowships on federal NIH grants T32 MH019524 and F32 MH114536 to D.C. The authors thank Alexander Ren for his assistance with data collection. We also thank Sarah Barber, Elizabeth Goldfarb, and Oded Bein for helpful comments on earlier versions of this manuscript.

## Author contributions

D.C. and L.D. conceptualized and designed the experiment; D.C. and C.G. collected data; D.C. and C.G. analyzed the data; D.C., C.G., and L.D. wrote the manuscript.

## Competing interests

The authors declare no competing interests.
