## [Peer Review File · Nature Communications]

Reviewers' comments:

Reviewer #1 (Remarks to the Author):

Summary: The present investigation explored how event boundaries influence characteristics of episodic memory, including temporal order, temporal distance, and source judgments. In 3 experiments, participants made indoor/outdoor judgments while encoding sequential lists of 32 pictures of everyday objects. During learning, an unrelated tone played in the left or right ear to indicate which hand should be used to make the indoor/outdoor judgment. The tone remained consistent for 8 trials before switching ears, forming an auditory event boundary. In Experiments 1 and 2, critical items appeared with two intervening items, while in Experiment 3, they appeared with 4 intervening items. Experiments 2 and 3 used eye-tracking, such that pupil size could be monitored as a neurophysiological arousal index. The behavioral performance was consistent across all experiments: Participants remembered items spanning event boundaries as having been further away, and they had poorer memory for temporal order when judging boundary-spanning, relative to same-context, pairs. Source memory (ear + item) was best for items immediately following a boundary. The pupillary data revealed that, although the pupils reliably dilated to all items, there were “spikes” following event boundaries. Using PCA and regression analyses with the three behavioral variables revealed that at least one pupil component was related to temporal order memory and temporal distance judgments. No effects were observed for source memory. In exploratory analyses examining arousal as a more constant state (via pupil size fluctuations throughout the task), lower trial-by-trial variability was related to better temporal estimation performance.

Comments: I read this paper with great interest, and was satisfied with the methods and analyses. I think that this is an important paper, and contributes meaningful new insights using a novel approach. I only have two comments for potential revisions.

Comment 1: I am reluctant to make this comment, because I loathe when it is made about my work, but the authors may want to include greater background on the neurophysiology and reliability of the pupillary reflex in the Intro (the Discussion does a great job of this). Although I consider the question of whether pupil size reliably indexes arousal and attention settled, there are many who are either unconvinced or uninformed, and may stop reading before getting to the Discussion.

Comment 2: It might help readers if the pupil components were described, even briefly, during the results section, so they know how to interpret the findings.

Reviewer #2 (Remarks to the Author):

In this manuscript by Clewett et al., entitled “Dynamic arousal signals construct memories of time and events”, the authors use pupillometry to investigate the relationship between arousal signals and event segmentation during memory formation. The research topic is interesting and timely, the experiments appear skillfully conducted, and the manuscript is very well written. Despite these positive remarks, I do have some concerns that make me hesitant whether the theoretical contribution of the manuscript is sufficient to warrant publication in Nature Communication.

The overall experimental paradigm has been used by the present research group in several studies before and indeed offers an elegant demonstration of how event boundaries contribute to the temporal structuring of episodic memories. In the present manuscript, the relationship between those established findings and variations in pupil size is investigated. Fluctuations in pupil size are used as a correlate of arousal changes, and the study aims to (1) determine whether event boundaries trigger robust spikes in arousal, and (2) whether such arousal changes are related to later mnemonic consequences of event segmentation. In my view, the first aim has already been addressed in the literature, and I do not think that the present study can make strong claims about the contribution of arousal states to event segmentation and later memory for time and events. The pupil response is interesting and may indeed track relevant mechanisms at play here, but it seems premature to conclude that it is arousal per se that mediates/promotes event segmentation and resulting changes in memory for the timing, order, and perceptual details. My main issues are elaborated below.

1) The pupil response is held to reflect increased arousal, mental effort—or, more generally, anything that ‘activates the mind’. Here, the authors link the pupil dilation response exclusively to arousal without clearly specifying what is triggering the increased arousal. There is an essential difference between more automatic reactions to salient stimuli (e.g., emotional content) and more voluntary allocation of attention (cognitive effort), and it is well established that the pupil size responds both to unexpected changes in the immediate environment (e.g., McCloy et al., 2017) and to more task-based uncertainty or difficulty (e.g., Nassar et al., 2012; Urai et al., 2017). Thus, it should be fundamental to specify the basis of the pupillary response to address questions about pupillary changes at event boundaries. Could we expect the pupil to remain unchanged when there is a change in a salient auditory tone stimulation? I think not. Thus, the first main finding that pupil size responds to event boundaries (as indicated by a change in tone) becomes trivial, and something that has been shown in the literature repeatedly. For example, Liao et al. (2016) reported the pupil size to increase in response to auditory stimuli changes (oddball) irrespective of whether attention was directed to the auditory input or not. It is then not surprising that a switch of attention to the other ear is sufficient to elicit a strong pupil dilation response (Fig 6b), but it doesn’t follow that this necessarily causes event-segmentation and later memory separation effects. A robust test of whether pupillary changes are related to the formation/updating of the current event model or instead to the registration of a tonal change seems to require a baseline comparison. To this end, it would have been informative to either manipulate the task-relevance of stimuli change (e.g., left/right ear: switch response hand; both ears: ignore tone) or

manipulate the emotional valence of the boundary markers (affecting arousal without affecting task-relevance). In summary, the current approach does not allow a proper characterization of the functional significance of the observed pupillary response.

2) The second main aim of the study is to examine the relationship between the pupil correlate of arousal and later memory separation effects. This step is necessary to infer anything about how pupil responses could be used to track event-segmentation. The authors use PCA to decompose the pupil response into multiple components and then correlate these with participants' memory for the timing, order, and perceptual details of the encoded events. In general, I believe that the functional significance of the different PCA components might not be as clear as the authors suggest when they write that they "enabled us to infer which mental processes (e.g., cognitive control, anticipation, etc.)" (p. 5). The authors should clarify this and further which criterion they used for deciding how many components to retain in the PCA.

A subset of the correlational analyses revealed significant relationships, but they are not entirely straightforward to interpret. The only component (4) that correlated with the subjective estimate of temporal distance explained only 2.66% of the variance in the pupil data. However, as there is no significant difference between boundary and non-boundary conditions for this pupil component, it is not readily apparent what this actually means. A significant difference between conditions seems like a prerequisite for conducting the correlational analysis at all. Also, the correlation plots suggest that there may be a few outlier data points that could be driving the reported effect. The other significant correlations involve an early and a later component (2 and 3) and temporal order memory. While this is potentially interesting, the meaning of those correlations remains unclear. The early component is likely to reflect some preparation, expectation, or anticipation of the tone onset. Thus, the result could simply reflect that participants who attend more to event boundaries are those who also display more event boundary effects during subsequent memory tests. The later component (2) overlaps nicely in time with previous findings, but the current approach doesn't allow a further specification of its functional significance. In addition, there is, in my view, no reason to see these effects only in experiment 3. If the pupil effect is indeed related to the formation/updating of event models, it should be present in both experiments. Similarly, if only arousal mediates event segmentation, one would have expected a relationship between the pupil correlate of arousal and the reliable source memory boost for boundary items. This suggests again that a further specification of the mediating factors of event-segmentation driven mnemonic phenomena is warranted. In summary, the correlational approach trying to link the pupil correlates of arousal to event segmentation, and later memory separation effects provided mixed results.

4) The authors may want to consider a 'subsequent memory' approach when relating pupil responses during memory formation with later memory separation effects. For example, given an adequate number of trials available, one could use linear discriminant analysis to classify pupil responses during boundary trials as a function of later memory judgments (within participant). Interestingly, when only examining boundary marker trials, would the pupil dilation response be predictive of later judgments of timing, order, and source memory?

5) It is not entirely clear in which temporal window the pupil signal was examined and baselined. From the information on pages 15, 19, and Fig 1 it appears that there is a fixation cross presented for 3 seconds before a stimulus. During this period, the onset of the tone occurs 1.5 seconds after the onset of the fixation cross and lasts for 1 second. The fixation cross is then on for another 500 ms before a picture is shown. The pupil is baselined in the period 500 ms before the tonal onset. This should thus give rise to a period of 1.5 seconds of baselined pupil data during the fixation cross. However, when looking into Figure 6b, 6c, and 6d, the analyzed baselined pupil signal comprises 3 seconds. It thus seems as if the last 1.5 seconds of this time window then represents 1.5 seconds during the succeeding picture presentation. I may, of course, have misunderstood the trial structure, but if the interpretation is correct, this means that the first and second half of the pupil data belong to two different visual inputs, which may severely affect the data and PCA. Please clarify.

6) I also think it would be valuable to look into pupil size in the baseline period. A growing literature within the adaptive gain theory, looking at exploration and exploitation control states of behavior has demonstrated that an exploratory mode is characterized by larger baseline pupil size (tonic changes) and an exploitative mode by phasic changes during the task period (e.g., Jepma et al., 2011). I think this could be quite revealing when examining the mechanisms underlying pupillary changes around event boundaries. Why not use the initial fixation cross in each trial for baseline correction? In this way, one would be able to track any anticipatory pupil response and disentangle this from tone-evoked responses.

7) The units for pupil diameter should be pixels or mm (Figures 6a and 6b).

8) Figure 6d is a bit confusing. It would be preferable to see the real factor scores and not the normalized scores (i.e., relative difference centered around 0). Otherwise, it is difficult to interpret the magnitude and direction of those effects.

Reviewer #3 (Remarks to the Author):

MS. 225824

Title: Dynamic arousal signals construct memories of time and events

Authors: Clewett, Gasser, & Davachi

Summary

This manuscript reports three experiments exploring how a surprising change in context or event boundary elicits pupil dilation, as well as changing in memory for perceptual qualities such as timing and order. Importantly, different pupillary response components were associated with different cognitive

components. For Experiment 1, people remembered objects after an event boundary worse than those not at an event boundary. Moreover, responses were consistent with the idea that event boundaries increased the perceived time duration across them, and relative temporal order memory was worse. Finally, memory for the ear of a tone heard at a boundary was worse than when the objects were not at an event boundary. Experiment 2 replicated Experiment 1 with the addition of pupillometry measurements. The basic results of Experiment 1 were replicated. Experiment 3 replicated Experiment 2 with the addition of a change in the actual temporal distance between items. The basic results of Experiments 1 and 2 were replicated. Moreover, in Experiments 2 and 3, the pupillometry data revealed increased pupil size at event boundaries, consistent with the idea that event boundaries are accompanied by increased arousal. The results are interpreted as showing that there is physiological arousal at event boundaries.

Recommendation

This is a well-written manuscript that explores interesting some ideas. However, I do have some points of concern that are listed below.

Major Points

1. The behavioral results reported here state that memory is worse for objects around or at an event boundary. However, other research shows the opposite (e.g., Swallow, Zacks, & Abrams, 2009). How can this be accounted for? Is it just the case that the tone switch serves as a distractor taking attention away from the main task?
2. The pupillometry results reported here suggest that memory is worse for objects around an event boundaries when arousal is greater. However, other research shows that increased arousal leads to better memory (e.g., LaBar & Phelps, 1998). How can this be accounted for?
3. Do the pupillometry data reflect event boundaries, or surprise at the tone shift? There is some evidence that surprise and event boundaries, while they may co-occur, do not always. In those cases, effects of the event boundaries and surprise can be separated out (Pettijohn & Radvansky, 2016). Given that it is difficult to do so with this data, can anything be concluded about the cause of the observed patterns of performance?

Response to Reviewers

Overarching Comments Regarding Major Changes

We thank the reviewers for their helpful comments and thoughtful suggestions. Before detailing our point-to-point responses to the reviewers' comments, we wanted to describe some major changes to the original manuscript. At the helpful suggestion of Reviewer 2 and the Editor, we have conducted several major re-analyses of the data. Before detailing our point-by-point responses to reviewers, we thought it would be helpful to first describe some of the major changes to the manuscript. At the end of this introductory section, we also describe data from a fourth experiment, which we intend to include in a forthcoming manuscript. We believe that these new results add significant confidence to our interpretations in the revised submission. For reasons that will become apparent, we have chosen not to include this new data, as it goes beyond the scope of the current manuscript. But we did include a summary of these new results in this response letter to further alleviate some of the concerns raised by Reviewer 2.

Brief overview of major additions

- 1) Revised **Figure 5** and **Figure 6** (pupil PCA and pupil-behavior correlations; in the original submission, these were **Figures 6 and 7**, respectively).
- 2) New, follow-up PCA's that bolster our interpretations regarding the functional significance of the pupil components (**Figure 6**).
- 3) We performed Spearman's rank correlations for the pupil-memory correlations to mitigate the influence of outliers.
- 4) **Figure 2** in the original manuscript has been moved to the Supplementary Materials (now **Figure S1**).
- 5) We included three new trial-level exploratory analyses in the **Supplementary Materials**:
 - a. **Supplementary Analysis 1**: Follow-up PCA's that target the tone-specific effects on pupil dilation (1.5s pupil-sampling window; Reviewer 2's request). This analysis was also used to make statistical comparisons between pupil component loadings between conditions during this specific time-window (**Figure S2**).
 - b. **Supplementary Analysis 2**: New trial-level linear mixed models examining the discrete effects of boundary-related tonic (pupil diameter at baseline) and phasic (pupil dilation to tone switches) arousal on subsequent memory (**Figure S3**). These analyses produced no significant effects, highlighting how simple, trial-averaging approaches to pupil analyses may obscure the more nuanced neural and cognitive mechanisms that segment memories at boundaries (identified by the PCA).

- c. **Supplementary Analysis 3:** New trial-level linear mixed models examining how the magnitude of the context shift (pitch changes between the pre-boundary and boundary tones) influences subsequent memory for boundary item pairs (**Figure S4**). This analysis helped us rule out potential effects of boundary salience on memory, lending further support to the idea that certain memory-related pupil components likely reflect task demands/relevance rather than sensory salience.

Summary of novelty and key contributions of this study

Conceptual novelty. We were pleased that the reviewers acknowledged the conceptual contributions of the reported work and how they lend further empirical support to our understanding of how arousal and context shapes the temporal structure of memory. The three studies reported in the original submission lay strong groundwork for understanding how different sub-components of pupil dilation may signal underlying arousal-related neuromodulatory processes that are known to influence episodic memory for simple stimulus associations (e.g., objects and their corresponding backgrounds).

While it is certainly well established that deviant auditory tones (e.g., oddballs) and salient tones elicit pupil dilation (e.g., Liao et al., 2016; Zekveld et al., 2018), very little work has examined the relationship between arousal and temporal memory organization; moreover, this small number of studies has primarily focused on how *stress or emotion* on a particular learning trial influences memory for trial-specific information. These studies are also limited, because they did not: (1) examine single-shot episodic memory encoding for temporally-extended sequences; (2) query subjective temporal distortions in episodic memory (e.g., subjective temporal distance); or (3) collect objective measures of arousal during encoding (e.g., pupil). All of these measures were collected/examined in the current study.

Additionally, our study examined arousal during *neutral experiences*, thereby highlighting that components of arousal influence cognition even when experiences are not explicitly considered emotional. Furthermore, still to date, relatively few studies examine how temporal associations are represented in memory more broadly, even though this rich sequential information is a hallmark feature of everyday memories. Although the notion that mental effort and/or cognitive control are engaged at boundaries to trigger segmentation has existed for more than ten years (Zacks et al., 2007), little is known regarding which cognitive processes are engaged as experiences unfold (e.g., perception/attention) and how they might differentially modulate various temporal memory outcomes. Our results contribute to filling this critical knowledge gap and bridge together largely independent literatures on event segmentation and memory organization.

We also underscore the very robust behavioral effects (encoding RT's and source/temporal memory) that replicated across three separate studies. While prior work from our lab and others shows that boundaries elicit an encoding RT slowing and three memory effects (source binding,

temporal order memory, and temporal distance ratings), this is the first study to independently modulate the context switch (boundary) from the presented memoranda.

For instance, the contextual stability is carried in tones that are embedded between the to-be-encoded images. In previous studies, boundaries, or shifts in context, were co-incident with the presented memoranda (for a review, see Clewett and Davachi, 2017). These prior boundaries included a switch in the perceptual category of items (e.g., faces to objects) and a specific judgement about those items (male/female vs. size, respectively; DuBrow and Davachi, 2013, 2016, 2017). In other studies, a colored border surrounded the objects and was not only relevant to the source features of the objects (i.e., participants rate pleasantness of object-color pairing), but the color shift defined the boundary itself (Heusser et al., 2017, 2018; also see Siefke et al., 2019).

The fact that these memory effects are very strong and reliable establishes this paradigm as an effective manipulation of event boundaries. Our study validates our paradigm in three separate experiments, thus establishing it as a useful tool for researchers who are interested in testing these ideas in the future. As described below, we also replicated these behavioral effects in a fourth experiment using this auditory event boundary manipulation. While this fourth dataset has not been included in the revised manuscript (more details below), we have included relevant results in this letter for the Reviewers and Editor to review.

Identifying novel associations between the pupil and temporal memory. As described in the original manuscript, a key advantage of using a PCA approach to analyze pupil data is that it offers a window into discrete arousal and cognitive processes that may be involved in segmenting memories (e.g., Johansson et al., 2018). First, we showed that components of pupil dilation previously linked to sympathetic and parasympathetic regulation are modulated by context shifts. We also linked these distinct pupil components to subjective and objective aspects of temporal memory, suggesting that potentially dissociable processes support different aspects of memory organization.

Reviewer 2 commented that the pupil dilations and memory effects co-occur and, hence, may not be causally related. We of course agree with that possibility, but also want to emphasize that the relationships are not simply a consequence of main effects that co-occur in both measures. These are correlations and thereby reflect discrete variability in both measures that relate to each other. This does not verify causality, but it does suggest that arousal processes may contribute to or at least signal underlying neuromodulatory processes (e.g., NE, ACh) that modulate these memory effects. Of course, future work will be needed to determine the necessity of arousal and/or different neuromodulators in memory separation or integration effects.

The issue of interpreting the functional significance of the components. A critical issue raised by Reviewer 2 and the Editor was that, in the current manipulation, it is somewhat difficult to identify the functional significance of the pupil components. This primarily stems from concerns that the boundary tones are both *salient* and *task relevant*, which makes it hard to

dissociate which of these factors are eliciting different sub-components of pupil dilation. Critically, however, we'd like to clarify important aspects of our design that may already help disentangle these two factors.

We agree with the reviewers that the salience of the boundary tones is indeed intimately related to the behavioral information the tones convey. The tone switches both deviate from the recent stream of auditory inputs (by switching pitch and ear) and cue a task-relevant motor response (switch hands). In our design, however, participants hear the same tones an equal number of times within a given list and also an equal number of times in each ear (16 per left ear and 16 per right ear per list). The only difference between a boundary tone and the same-context tone is: (a) whether it switched to the other ear from the immediately preceding trial, and (b) what this switch means for ongoing behavior, or its task relevance (i.e., cue to switch hands). Thus, we believe that it is valid to interpret our pupil dilation effects as being driven by the task relevance of the boundary tones and not the sensory features/salience of the tone onset itself, given that the latter is equivalent across conditions.

To explore the functional significance of the components further, we performed separate follow-up 3s-window PCA's on the boundary trials and same-context trials (**Figure 6a,b**). Prior work shows that motor responses elicit pupil dilations that peak 600-900ms later (Beatty, 1982; Steinhauer 1992; Richer et al., 1983; Shiga and Ohkubo, 1979), which is consistent with the peak timing of component #4 (early-peaking; 800ms). Our design offered an opportunity to test this component: across the two conditions (boundary and same-context), there were four timepoints where a stimulus came onscreen (two tones and two images), and only one of those timepoints — the same-context tone — did *not* require any sort of re-mapping of the motor response (e.g., switching hands or judging objects as indoor/outdoor via button press; **Figure 6c**). Consistent with a motor-response interpretation for the early-peaking component, the same-context tone was the only tone-/image-related timepoint when loading on this component (#4) was not evident (**Figure 6c**). Thus, our pupil-memory correlation results suggest that individuals who show more of this arousal response to a boundary, which may be indicative of the re-mapping of a task/motor response, also seem to remember recent events as having occurred farther apart in time. For convenience, this plot is displayed below. The results are described on page 10 of the Results section.

Figure 6. Temporal characteristics of pupil dilation evoked by tones and their subsequent images. (a) A temporal principal component analysis (PCA) on the pupil dilation data revealed four pupil components that had distinct shapes across time. (b) Two follow-up PCAs separated by condition help illustrate how loading on these components differed between event boundary trials and same-context trials. Vertical dashed lines signify the onsets of the tones and their subsequent images. In both conditions, most of the temporal characteristics of the pupil component loadings were qualitatively similar, except for the early-peaking component (component #4; turquoise). (c) To better illustrate these differences, only component four's loading time-course is displayed. The plot reveals evidence of this early-peaking component in response to the boundary tones, boundary images, and same-context images (red arrows). However, this component did not peak in response to same-context tones (red shaded area), which was the only stimulus type (tone or image) that did not require a motor response. (d) Only the slowly-decreasing component (#3; sky blue) is highlighted to illustrate potential differences in its loading patterns over time. A peak in this component's loadings is identifiable for each tone and image type (red arrows).

Next, to quantify this difference in the early-peaking component's loadings between conditions, we limited the time-window of the PCA to 1.5s (which was also suggested by Reviewer 2). This enabled us to look at boundary versus same-context effects on pupil loadings for the tones specifically (i.e., this time-window does not include pupil responses to the onset of the images that followed each tone; **Supplementary Analysis #1**). The results confirmed that, compared to

same-context tones, boundary tones significantly enhanced loadings on the early-peaking component (**Figure S2**). For convenience, these plots are displayed below. The results of this analysis are briefly reported in the Results section on page 11. Additional details of this analysis can be found in **Supplementary Analysis #1** on pages 1-2 of the **Supplementary Materials**.

Supplementary Figure 2. (a) Temporal features of tone-evoked pupil dilation identified by a temporal principal component analysis (PCA) focused on the 1.5s tone period. For illustrative purposes, separate PCAs were performed on the boundary and same-context trials to show qualitative differences in pupil loadings across time. Vertical dashed lines signify the onsets of the tones and their subsequent images. The 1.5s PCA revealed three significant features of pupil dilation that had distinct shapes over time and matched three of the components identified in the 3-s window PCA. Component loadings reflect “raw” values from the rotated solution, so are on the same scale as the original pupil inputs. (b) Condition-related differences on pupil component loadings. The boundary tones (blue colors) versus same-context tones (gray) differentially modulated loading scores for component #1 (intermediate) and component #3 (early-peaking), but not for component #2 (slowly-decreasing). All error bars represent standard error of the mean (SEM). *** $p < .001$; ** $p < .01$; ~ $p < .10$.

We believe these new analyses will be very informative for readers, and also add confidence to our interpretations regarding the motor/decision processes captured by the early-peaking

component that was related to subjective temporal memory. We interpret this result as shedding light on the nature of theoretical ‘event models’ proposed by Zacks et al. (2007) on page 15 of the Discussion:

“Our pupil-memory correlation results demonstrate that individuals who show more of this arousal response to a boundary, which may be indicative of greater re-mapping of a motor response, also seem to remember recent events as having occurred farther apart in time. One possible interpretation of this effect is that people are constantly maintaining an active representation of the world, or ‘event model’, which requires sustained cognitive processing [8]. At an event boundary, even greater cognitive control or sustained processing may be necessary to update this event model with new contextual information. This cognitive-updating process, in turn, may shift our memory systems to prioritize processing of a new context, creating greater separation in memory space. In the current task, motor-response mapping provides essential, task-relevant contextual information that helps to distinguish one event from the next. In so doing, it also provides a shared mental context for binding successive items to a common memory representation (e.g., “left hand/sided event”).”

In summary, we believe that the functional significance of the two memory-related pupil components is actually relatively clear. The early-peaking component (associated with subjective temporal distance ratings) only appeared when a motor response was initiated/required, suggesting it relates to motor re-mapping. The slowly-decreasing component (associated with temporal order memory) was triggered prior to the onset of a tone. Given this specific timing, we believe it is reasonable to conclude that this signifies some form of response anticipation. Importantly, both of these interpretations are also supported by prior work using PCA and behavioral paradigms.

It is noteworthy that, unlike the 3-s PCA that accounted for the combined pupil responses to tones and images, the tone switches alone (1.5-s window) did not modulate loadings on the slowly-decreasing, ‘anticipatory’ component. This result suggests that pupil-linked anticipation following a tone switch is present to a similar degree for both ‘boundary’ images and images from the same-context (for comparison, see **Figure S2b** and **Figure 5d**). We interpret this result as being consistent with evidence suggesting that both the context shift and subsequent boundary items are processed differently than same-context information, with anticipation of both being important for the segmentation of subsequent memory.

We also note that the intermediate component (#2; royal blue), whose functional significance is arguably the hardest to interpret in our design, was no longer significantly related to temporal order memory when we applied Spearman’s rank correlation analyses to control for the effects of outliers (suggested by Reviewer 2 and noted below). Thus, any interpretation regarding the significance of this component is no longer warranted.

Stressing the importance of PCA approach to pupil analyses. Per Reviewer 2’s request, we’ve now included trial-level linear mixed model analyses to examine the relationship between

tonic (pupil diameter during the pre-boundary baseline period) and phasic (average tone-evoked pupil dilation) arousal and memory (see **Supplementary Analysis #2**). The results revealed no significant effects between these pupil measures and memory performance across boundaries. This demonstrates that the more nuanced cognitive and arousal mechanisms detected by the PCA would have been obscured by more traditional pupil-averaging approaches. Based on the existing literature on how these pupil components relate to different aspects of behavior and autonomic pathways, our data provide new insight into which cognitive and neural processes may be involved in segmenting memories at boundaries. The results of these trial-level analyses are displayed below (**Figure S3**). They are briefly mentioned on pages 12-13 of the Results section. Details of this analysis can be found on page 2 of the **Supplementary Materials**.

Supplementary Figure 3. The relationship between trial-level pupil measures of tonic and phasic arousal and subsequent memory broken down by eye-tracking experiment. (a) To assess global, or tonic, levels of arousal just prior to a boundary, pupil diameter values were averaged across the 500ms baseline period preceding a tone switch. Mixed effect linear modeling revealed no significant associations between baseline pupil diameter and any of the three memory outcomes. (b) To assess stimulus-evoked, or phasic, levels of arousal induced by boundaries, pupil diameter

values were averaged across the window 1-2s after the onset of tone switch. These values were then baseline-normalized by subtracting the average pupil diameter in the 500ms prior to the onset of the tone switch. Mixed effect linear modeling revealed no significant associations between tone-evoked pupil dilation and any of the three memory outcomes.

Examining how the magnitude of the context shifts impact subsequent memory. As mentioned above, the subtractions we performed in the pupil component-memory correlations help to control for any lower-level sensory salience of the tones. Even so, we agree that the salience of the boundary tones is intimately related to – and largely driven by – their relevance to behavior. To rule out the potential effects of lower-level salience on memory even further, we have now added an additional trial-level analysis (see **Supplementary Analysis #3**).

In this mixed linear model, we examined whether the sensory salience of the tone switches was related to pupil dilation and subsequent memory. To test this possibility, we leveraged the varying magnitude in tone pitches change across boundaries (ranging from 100Hz to 500Hz), which provided a more sensitive measure of trial-by-trial changes in boundary salience. The results revealed no significant association between the magnitude of tone pitch changes from pre-boundary to boundary tones and pupil dilation at those boundaries. There also was no significant relationship between tone pitch changes at boundaries and memory outcomes at those boundaries, suggesting that the change in the sensory properties of the tone itself did not elicit different levels of phasic arousal or modulate episodic memory (**Figure S4b**). Rather, the behavioral significance, or task relevance, of the tone switches appeared to be sufficient to elicit memory separation irrespective of salience. For convenience, these results are displayed in the figure below. In the revision, the results are briefly mentioned on page 13 of the Results section of the main text. The details of this analysis are reported on page 3 of the **Supplementary Materials**.

a) Effects of absolute pitch change at boundaries

Supplementary Figure 4. Relationships between the magnitude of a tone-related context shift, subsequent memory, and tone-evoked pupil dilation. (a) The magnitude of a tone-related context shift was quantified by measuring the absolute change in the pitch (frequency) of the tones for the pre-boundary tone to the boundary tone. Mixed effect linear modeling revealed no significant association between the pitch change at boundaries and tone-evoked pupil dilation. (b) In addition, changes in tone pitch at boundaries did not correlate with any of the three memory outcomes. Hz = Hertz.

Key replication of behavioral and eye-tracking results in a fourth experiment. Importantly, we replicated the temporal memory effects and key eye-tracking results in a fourth experiment, too. When we received the decision on this manuscript, we had already begun collecting data for another auditory event boundary study. This fMRI/eye-tracking study used the same type of auditory event boundary manipulation, but with a few modifications. A total of thirty-two participants in this study (12 male; $M_{\text{age}} = 21.63$, $M_{\text{SD}} = 2.71$) had analyzable behavioral data, and twenty-eight participants had usable eye-tracking and behavioral data (11 male; $M_{\text{age}} = 21.71$, $M_{\text{SD}} = 2.83$).

This new experiment was designed to address many of the novel questions raised in our original submission. We had also included an important change to the original design of the experiment, which involved manipulating the predictability of stimulus timings. This new dataset thereby provided a unique opportunity to evaluate the functional significance of the pupil

components more directly. Unfortunately, we found that incorporating this extra experiment was impractical due to length/word constraints, and we believe that the new results are beyond the scope of the current manuscript (issues discussed further below). However, here we explain the key findings that we replicated to increase confidence in our original results.

The goal of this new experiment was to target one of the key findings in the original manuscript: Does predictability shape the temporal structure of memory? This idea was motivated by our finding that one of the pupil components that likely signifies response anticipation (component #3; slowly-decreasing component) was related to impaired order memory (i.e., event segmentation) for information spanning a boundary in Experiment 3 (see **Figure 7b** in revised manuscript).

To obtain a better understanding of the functional significance of this component (mainly with respect to anticipation), we varied the predictability of the stimulus timings in the auditory event boundary task; that is, we pseudorandomized the timing of the tones and images to be 3, 5 or 7 seconds apart; we also controlled for the objective temporal distance between to-be-tested item pairs to be the same. Thus, while the overall event structure was still consistent (i.e., tone changes after every 8 items), the exact stimulus timings were less predictable. This design provided an excellent opportunity for us to follow up on a key pupil-behavioral finding in the original manuscript, suggesting that the predictability of event boundaries (and therefore top-down cognitive processes) may drive event segmentation in memory.

Our new results support this hypothesis and replicate/validate many of our original findings. There were also new findings that expand upon the results reported in the original submission. Specifically, we found that:

- (1) We once again found that the tone switches, or event boundaries, elicited a significant pupil dilation. The results revealed that boundary tones, $t(27) = 6.48$, $p < .001$ [CI: 69.90, 134.66], but not same-context tones, $t(27) = -.71$, $p = .48$ [CI: -17.42, 8.43], elicited significantly greater pupil dilation than baseline (Figure 6c). Boundary tones also evoked significantly greater pupil dilations than same-context tones, $t(27) = 7.03$, $p < .001$ [CI: 75.63, 137.92]. Together, these results demonstrate that auditory event boundaries were indeed salient and triggered a momentary increase in physiological arousal.
- (2) The three prototypical pupil PCA components identified in the 1.5s PCA in the revised manuscript (see **Supplementary Analysis #1**) replicated in a new dataset. This replication points to the robustness of these pupil components and adds to our confidence that these are stable neural pathways that regulate pupil dilation (e.g., Steinhauer and Hakerem, 1992). Of note, due to timing differences with the pseudo-randomized ISI's in the new experiment, we could not examine the same 3s pupil-sampling PCA window as the original paper. In short, 2/3rds of the trials in the new experiment did not encompass both the tones and their subsequent images (the time gap would only be the same as the 1.5s tone-to-image gap in the original experiments) on the 3s ISI trials. Thus, we had too few trials to include the new dataset in a 3s PCA

collapsed across all datasets. Nevertheless, the replication of the 1.5s PCA window adds confidence to the idea that tones elicit a pupil dilation with stable, temporally-distinct patterns.

- (3) The previously observed relationship between boundary-related loadings on this anticipatory component (component #3) and boundary-related effects on temporal order memory was no longer significant. This reinforces the idea that predictability is important for event segmentation, as signaled by distinct, anticipatory characteristics of pupil dilation at boundaries. Top-down attentional processes might therefore play an important role in proactively segmenting or chunking contextually-related information into discrete mnemonic events.
- (4) Remarkably, the behavioral results from this new study also replicate the key findings in the original manuscript. Specifically, the participants: (1) were slower to make indoor/outdoor judgements for boundary items compared to items from a stable auditory context, $t(31) = 4.71$, $p < .001$ [C: 36.41, 92.10]; (2) remembered boundary-spanning pairs as having appeared farther apart in time than same-context pairs, $t(31) = 2.26$, $p = .031$, [CI: 0.0073, 0.14]; and (3) were worse at remembering the temporal order of boundary-spanning item pairs relative to same-context pairs, $t(31) = -3.91$, $p < .001$, [CI: -0.10, -0.033]. Notably, source memory was not tested in this fourth study so that we could focus on temporal memory effects.
- (5) The stability of arousal levels over more prolonged periods of time (reported in **Supplementary Analysis #4** in the current revised manuscript) again predicted worse temporal memory in the new experiment ($z = -2.08$, $p = .037$). Additionally, consistent with the pattern observed in Experiments 2 and 3, trial-level changes in arousal stability also showed a statistical trend towards predicting higher ratings of temporal distance between item pairs ($t = 1.25$, $p = .21$). Thus, this relationship showed the same positive directionality as the original two eye-tracking experiments.

Taken together, these new findings suggest that the predictability of event boundaries may help shape the temporal organization of events in memory, with more predictable boundaries leading to greater event-parsing in episodic memory. This dataset bolsters our original results, and provides more information about the functional significance of different pupil dilation components and their relationship with temporal memory outcomes. Unfortunately, describing all of the methodological changes and new results/interpretations is somewhat impractical in terms of integrating all of these studies into one manuscript (increases the word count by over 3,000 words). More importantly, we believe that the findings from the three original studies make novel and important contributions to the fields of cognitive neuroscience and learning and memory. Many of the new questions inspired by the original manuscript will be addressed in the planned, forthcoming fMRI/pupil/behavioral manuscript, which builds upon the strong theoretical foundation laid in the original submission.

Point-By-Point Response to Reviewers

****NOTE**** We have sub-divided some of the reviewers' comments into sub-sections to clarify our responses to some of the complex, multi-part questions.

Reviewer #1 (Remarks to the Author):

Summary: The present investigation explored how event boundaries influence characteristics of episodic memory, including temporal order, temporal distance, and source judgments. In 3 experiments, participants made indoor/outdoor judgments while encoding sequential lists of 32 pictures of everyday objects. During learning, an unrelated tone played in the left or right ear to indicate which hand should be used to make the indoor/outdoor judgment. The tone remained consistent for 8 trials before switching ears, forming an auditory event boundary. In Experiments 1 and 2, critical items appeared with two intervening items, while in Experiment 3, they appeared with 4 intervening items. Experiments 2 and 3 used eye-tracking, such that pupil size could be monitored as a neurophysiological arousal index. The behavioral performance was consistent across all experiments: Participants remembered items spanning event boundaries as having been further away, and they had poorer memory for temporal order when judging boundary-spanning, relative to same-context, pairs. Source memory (ear + item) was best for items immediately following a boundary. The pupillary data revealed that, although the pupils reliably dilated to all items, there were "spikes" following event boundaries. Using PCA and regression analyses with the three behavioral variables revealed that at least one pupil component was related to temporal order memory and temporal distance judgments. No effects were observed for source memory. In exploratory analyses examining arousal as a more constant state (via pupil size fluctuations throughout the task), lower trial-by-trial variability was related to better temporal estimation performance.

I read this paper with great interest, and was satisfied with the methods and analyses. I think that this is an important paper, and contributes meaningful new insights using a novel approach. I only have two comments for potential revisions.

Comment 1: *I am reluctant to make this comment, because I loathe when it is made about my work, but the authors may want to include greater background on the neurophysiology and reliability of the pupillary reflex in the Intro (the Discussion does a great job of this). Although I consider the question of whether pupil size reliably indexes arousal and attention settled, there are many who are either unconvinced or uninformed, and may stop reading before getting to the Discussion.*

RESPONSE: This is an excellent suggestion and we believe that making this change has improved the framing of the overall paper. We have now included the following text in the Introduction on pages 5-6:

"The primary aim of this study was to determine if boundary-induced arousal responses relate to later memory separation effects. Thus, in

Experiments 2 and 3, we also measured pupil diameter continuously throughout our sequence-learning task to index changes in autonomic arousal across time [42, 43]. Increasing evidence also shows that pupil diameter may be a reliable marker of cognitive processing [44-47], and thereby provides a potential indirect measure of the underlying mental processes that organize experience into discrete memories. To specifically examine the relationship between boundary tone-evoked pupil dilation and memory, we adopted a temporal principal component analysis (PCA). In previous work, this analysis technique has been used to dissociate temporal characteristics of pupil dilation that may reflect different cognitive processes [48-54]. For instance, one recent study applied PCA to show that a specific sub-component of pupil dilation is triggered just prior to the onset of memory retrieval decision, suggesting that this component signals the anticipation of an impending decision/response (Johansson et al., 2018).

Temporal PCA has also been combined with studies manipulating different lighting conditions to distinguish overlapping contributions of parasympathetic and sympathetic autonomic pathways to pupil dilation. Oddball stimuli have been shown to modulate an early-peaking (~800ms) component under moderate light, but not under darker conditions when parasympathetic tone is minimal [52, 53]. This finding suggests that this early-peaking aspect of pupil dilation may specifically index parasympathetic inhibition, which elicits pupil dilation via relaxation of the sphincter muscle [55]. With respect to behavior, this early-peaking pupil component is increasingly engaged by cognitive load [52] and peaks within a time-window consistent with motor responses (600-900ms; [51, 56]). Taken together, PCA holds important advantages over more conventional pupil-averaging analyses, because it offers a unique window into different mental (e.g., anticipation, mental effort, motor responses etc.) and neural processes that may be engaged by event boundaries.”

Comment 2: *It might help readers if the pupil components were described, even briefly, during the results section, so they know how to interpret the findings.*

RESPONSE: We thank Reviewer 1 for this helpful suggestion. To contextualize our pupil results within the broader literature, we have now added short descriptions of the components at the end of each results section.

Reviewer #2 (Remarks to the Author):

Summary: In this manuscript by Clewett et al., entitled “Dynamic arousal signals construct memories of time and events”, the authors use pupillometry to investigate the relationship between arousal signals and event segmentation during memory formation. The research topic is interesting and timely, the experiments appear skillfully conducted, and the manuscript is very well written. Despite these positive remarks, I do have some concerns that make me hesitant whether the theoretical contribution of the manuscript is sufficient to warrant publication in Nature Communication.

The overall experimental paradigm has been used by the present research group in several studies before and indeed offers an elegant demonstration of how event boundaries contribute

to the temporal structuring of episodic memories. In the present manuscript, the relationship between those established findings and variations in pupil size is investigated. Fluctuations in pupil size are used as a correlate of arousal changes, and the study aims to (1) determine whether event boundaries trigger robust spikes in arousal, and (2) whether such arousal changes are related to later mnemonic consequences of event segmentation. In my view, the first aim has already been addressed in the literature, and I do not think that the present study can make strong claims about the contribution of arousal states to event segmentation and later memory for time and events. The pupil response is interesting and may indeed track relevant mechanisms at play here, but it seems premature to conclude that it is arousal per se that mediates/promotes event segmentation and resulting changes in memory for the timing, order, and perceptual details. My main issues are elaborated below.

Comment 1A: *The pupil response is held to reflect increased arousal, mental effort—or, more generally, anything that ‘activates the mind’. Here, the authors link the pupil dilation response exclusively to arousal without clearly specifying what is triggering the increased arousal. There is an essential difference between more automatic reactions to salient stimuli (e.g., emotional content) and more voluntary allocation of attention (cognitive effort), and it is well established that the pupil size responds both to unexpected changes in the immediate environment (e.g., McCloy et al., 2017) and to more task-based uncertainty or difficulty (e.g., Nassar et al., 2012; Urai et al., 2017). Thus, it should be fundamental to specify the basis of the pupillary response to address questions about pupillary changes at event boundaries. Could we expect the pupil to remain unchanged when there is a change in a salient auditory tone stimulation? I think not. Thus, the first main finding that pupil size responds to event boundaries (as indicated by a change in tone) becomes trivial, and something that has been shown in the literature repeatedly.*

RESPONSE: This is an excellent point, and we appreciate Reviewer 2 drawing additional attention to those papers and to the framing of our results. We completely agree that, given the current literature, we had every expectation to see a main effect of pupil dilation at a boundary (here, defined as a behaviorally relevant tone change). And we agree that merely showing that result would not warrant sufficient novel interest. However, the paper further dissociates the pupil dilation into different (functional) components, some of which relate to temporal memory and others which do not. Thus, the novelty in our paper arises from the relationship between pupil components and differential memory behavior and suggests that dynamic increases in pupil dilation at boundary as well as temporal stability in pupil components may contribute to memory separation and integration.

To make it clear that this first finding (pupil dilation at event boundaries) is indeed predicted and expected based on prior work, we now include a short description of prior research that has examined the effects of event structure on pupil dilation. We also make explicit connections between this work and our memory-related hypotheses to underscore the goals/novelty of the present study. These descriptions can be found in the Introduction section on page 4:

“Of relevance to the current study, emerging evidence also suggests that arousal states may be sensitive to the structure of temporally-extended events [40, 41]. For instance, pupil dilation occurs when a highly-organized and repeated sequence of auditory tones transitions to a randomized sequence tones, but not during the opposite transition [41]. This suggests that pupil-linked arousal can modulate and/or signal context shifts in a manner consistent with the presence of event boundaries. Critically, however, it remains unclear whether these dynamic arousal responses also affect subsequent memory.”

We also elaborate on prior findings that have linked various types of salient stimuli to pupil dilation in our response to Reviewer 1 Comment #1. We'd also like to clarify that we are not arguing that *only* arousal is involved in facilitating event segmentation. Rather, arousal is one contributing factor (or index of underlying cognitive processes) that is likely interacting with (or signaling) other complex memory and attention processes.

Comment 1B: *For example, Liao et al. (2016) reported the pupil size to increase in response to auditory stimuli changes (oddball) irrespective of whether attention was directed to the auditory input or not. It is then not surprising that a switch of attention to the other ear is sufficient to elicit a strong pupil dilation response (Fig 6b), but it doesn't follow that this necessarily causes event-segmentation and later memory separation effects. A robust test of whether pupillary changes are related to the formation/updating of the current event model or instead to the registration of a tonal change seems to require a baseline comparison. To this end, it would have been informative to either manipulate the task-relevance of stimuli change (e.g., left/right ear: switch response hand; both ears: ignore tone) or manipulate the emotional valence of the boundary markers (affecting arousal without affecting task-relevance). In summary, the current approach does not allow a proper characterization of the functional significance of the observed pupillary response.*

RESPONSE: We appreciate Reviewer 2's excellent suggestions and cogent argument. We agree that altering the manipulation to test whether certain aspects of top-down attention (e.g., task relevance, expectation etc.) at the event boundaries could be used to shed light on the significance of pupil dilation and the cognitive processes that mediate the influence of boundaries on later temporal memory. We also agree that further work should utilize a design such as the one you suggest to disentangle arousal to unexpected, irrelevant (oddball) auditory tones from those where the tone signals a task switch (as in our current design). We did not include that manipulation in this experiment, because we were and remain primarily interested in testing how and whether pupillary components relate to temporal (and source) memory behavior. The addition of a baseline auditory tone that did not signify any sort of task change may or may not have produced event segmentation in memory; thus, we would not have known whether we had sufficient power in our design to properly assess memory behavior.

Given those caveats, however, we want emphasize that there are important aspects of our design that already help disentangle task relevance from the salience of the tone switches (event boundaries). The salience of these tones lies in their significance/task-relevance, which makes this task different than the series of experiments conducted by Liao et al. (2016). By

performing subtractions between the boundary and same-context conditions, we are effectively controlling for lower-level sensory properties of the stimuli. In our design, participants are hearing the same tones an equal number of times within a given list (8 of each tone type per event). They are also hearing an equal number of tones in each ear (16 per left ear and 16 per right ear per list). The only difference between a boundary tone and the same-context tone is: (a) whether it switched to the other ear from the immediately preceding trial, and (b) what this switch means for ongoing behavior, or its task relevance (i.e., cue to switch hands). Thus, we believe that it is valid to interpret our pupil dilation effects as being driven by the task relevance of the boundary tones and not the sensory features/salience of the tone onset itself, given that the latter is equivalent across conditions.

Because of this, our experiment differs from the largely salience-driven effects of white noise versus pure tone oddball stimuli and standard stimuli used in Liao et al. (2016). In other words, in the study by Liao et al. (2016), the white noise bursts were highly arousing and salient and deviate from the surrounding 'standard' sounds, whereas our tones don't strictly differ in overall sensory salience. Furthermore, our data don't necessarily fall under the classic definition/manipulation of 'oddballs', per se, because these sequences are regularly structured (i.e., the tone always switches every 8 items, and is highly predictable). In the general oddball paradigm, temporal distance between oddballs is usually randomized to make them highly unpredictable and surprising. Deviant oddball stimuli are also usually defined by being physically different from standard stimuli. Our tone switches (boundaries) don't fully satisfy either of these criteria for a classic oddball effect.

Nevertheless, to further increase the interpretability of the pupil components, we characterized and plotted the dynamic fluctuations of each component with respect to the tone onset and the following image onset. We describe these methods and results in more detail in our next responses (Reviewer 2 Comment #2A).

While we like the suggestion of manipulating emotional valence of the boundary markers – and are in fact running this study in our lab right now – this fails to address the main goals of our study. Much of the work on how arousal influences episodic memory focuses on emotional stimuli. The goal of this paper was to demonstrate that arousal elicited by event boundaries may be a core mechanism of *everyday memory organization*. As such, we would expect to see a relationship between arousal/pupil response and temporal memory under 'neutral' conditions (or those not explicitly considered emotionally salient). Including emotional stimuli might also obscure other cognitive processes that are engaged by boundaries to elicit event segmentation in memory, as emotional stimuli have complex semantics, affective significance, and may trigger lingering ruminative processes. We have predictions about how these boundaries may influence memory organization, but these ideas aren't relevant to the current manuscript.

Comment 2A: *The second main aim of the study is to examine the relationship between the pupil correlate of arousal and later memory separation effects. This step is necessary to infer anything about how pupil responses could be used to track event-segmentation. The authors use PCA to decompose the pupil response into multiple components and then correlate these*

with participants' memory for the timing, order, and perceptual details of the encoded events. In general, I believe that the functional significance of the different PCA components might not be as clear as the authors suggest when they write that they "enabled us to infer which mental processes (e.g., cognitive control, anticipation, etc.)" (p. 5). The authors should clarify this and further which criterion they used for deciding how many components to retain in the PCA.

RESPONSE: In response to the last point, we now clarify that we used Kaiser criterion (Kaiser, 1960) as a cut-off for detecting/keeping significant pupil components from the PCA.

We agree that we do not know for sure what underlying function each pupil component supports and had added text to the new manuscript to highlight that these are inferences made from previously published studies (page 16 of Discussion section):

"Importantly, some of our interpretations concerning the functional significance of these pupil components are drawn from previous studies using PCA."

However, motivated by this comment, we have now conducted additional analyses to characterize each pupil component in more depth using the current data. Specifically, we have now conducted *separate* exploratory PCAs for boundaries and same-context trials in order to illustrate differences in the behavior/features of these component loadings over the course of each trial (see **Figure 6b**). These analyses can be found in the Results section on pages 10-11.

Figure 6. Temporal characteristics of pupil dilation evoked by tones and their subsequent images. (a) A temporal principal component analysis (PCA) on the pupil dilation data revealed four pupil components that had distinct shapes across time. (b) Two follow-up PCAs separated by condition help illustrate how loading on these components differed between event boundary trials and same-context trials. Vertical dashed lines signify the onsets of the tones and their subsequent images. In both conditions, most of the temporal characteristics of the pupil component loadings were qualitatively similar, except for the early-peaking component (component #4; turquoise). (c) To better illustrate these differences, only component four's loading time-course is displayed. The plot reveals evidence of this early-peaking component in response to the boundary tones, boundary images, and same-context images (red arrows). However, this component did not peak in response to same-context tones (red shaded area), which was the only stimulus type (tone or image) that did not require a motor response. (d) Only the slowly-decreasing component (#3; sky blue) is highlighted to illustrate potential differences in its loading patterns over time. A peak in this component's loadings is identifiable for each tone and image type (red arrows).

What is also critical to note is that this early-peaking component (#4) relates to temporal distance ratings, suggesting that motor-mapping relates to how memories become segmented later on (**Figure 7a**). Thus, the novel finding here is not necessarily that this pupil component signals a motor response, but rather that a change in the motor response mapping ('re-mapping' of mental/task context) may nudge context representations just enough to influence how the

ensuing representations are grouped together in memory. This distinction is critical. We have added the following text to the Discussion on page 15:

“Our pupil-memory correlation result demonstrates that individuals who show more of this arousal response to a boundary, which may be indicative of greater re-mapping of a motor response, also seem to remember recent events as having occurred farther apart in time. One possible interpretation of this effect is that people are constantly maintaining an active representation of the world, or ‘event model’, which requires sustained cognitive processing [8]. At an event boundary, even greater cognitive control or sustained processing may be necessary to update this event model with new contextual information. This cognitive-updating process, in turn, may shift our memory systems to prioritize processing of a new context, creating greater separation in memory space. In the current task, motor-response mapping provides essential, task-relevant contextual information that helps to distinguish one event from the next. In so doing, it also provides a shared mental context for binding successive items to a common memory representation (e.g., “left hand/sided event”).”

Comment 2B: *A subset of the correlational analyses revealed significant relationships, but they are not entirely straightforward to interpret. The only component (4) that correlated with the subjective estimate of temporal distance explained only 2.66% of the variance in the pupil data. However, as there is no significant difference between boundary and non-boundary conditions for this pupil component, it is not readily apparent what this actually means. A significant difference between conditions seems like a prerequisite for conducting the correlational analysis at all.*

RESPONSE: Although this is a valid point, we don’t believe that performing the correlation analyses are predicated on finding significant main effects of boundaries on pupil loadings. As we argued in the original paper, it may be the case that some pupil components reflect more generalized cognitive processes that facilitate information processing as events unfold. What’s perhaps more important is that our main correlation analyses are leveraging individual differences in behavior. Thus, our goal was to identify pupil-memory associations that could account for variability in boundary-related arousal and memory difference across participants.

We also note that this idea/approach is analogous to studies that directly integrate behavior into their neural analyses. For instance, a subsequent memory analysis using fMRI is used to model BOLD signal patterns that are elicited, say, by items that are later remembered versus items that are later forgotten. This analysis can still be performed in the absence of a main effect of memory performance in behavior. What the results of these types of analyses reveal is that certain brain activation patterns occur *when* people successfully encode information, even though there were no observable differences in memory performance on average. We view our pupil PCA/correlation approach as being similar to this, particularly given that there were individual differences in our pupil and memory measures.

Comment 2C: *Also, the correlation plots suggest that there may be a few outlier data points that could be driving the reported effect.*

RESPONSE: We thank Reviewer 2 for pointing this out. To control for potential outlier effects, we now perform Spearman's rank correlation coefficient correlations, a non-parametric regression technique that is robust to the influence of outliers. The new correlation results replicated much of the results initially reported in the prior submission (see **Figure 7**). We still find a significant relationship between loading on the slowly-decreasing component (component #3) and impaired order memory across boundaries ($\rho = -0.36$, $p = .048$). We found that individuals who exhibited a time expansion effect in memory for boundary-spanning pairs also showed more evidence of the early-peaking pupil component during boundaries versus non-boundaries ($\rho = 0.25$, $p = .047$). The relationship between the early-peaking pupil component and temporal distance memory was significant in Experiment 3 ($\rho = 0.43$; $p = .016$) and showed the same positive pattern in Experiment 2 ($\rho = 0.21$; $p = .22$) when these experiments were analyzed separately. The fact that the direction and magnitude of these correlations was similar across studies helps to explain why there was a significant main effect when the data were collapsed across studies. The only exception to the original results is that, when using this more robust technique, we no longer find a significant association between temporal order memory and loading on the intermediate component (component #2) in Experiment 3.

Comment 2D: *The other significant correlations involve an early and a later component (2 and 3) and temporal order memory. While this is potentially interesting, the meaning of those correlations remains unclear. The early component is likely to reflect some preparation, expectation, or anticipation of the tone onset. Thus, the result could simply reflect that participants who attend more to event boundaries are those who also display more event boundary effects during subsequent memory tests.*

RESPONSE: The possibility that our pupil-memory correlation for component #3 reflects differences in attention to boundaries is definitely a valid possibility, and one that we consider interesting. In fact, we believe this individual-differences effect offers strong support for our hypotheses: those individuals who proactively attend to boundaries to a larger degree are also those who show greater event segmentation effects in episodic memory.

Attention is indeed part and parcel of an event boundary. Leading theories of event segmentation (e.g., Radvansky, 2012; Zacks et al., 2007) posit that boundaries shift attention to relevant new boundary information and toward the processing of new contextually-relevant information. Across many studies, this attention effect has been illustrated via the slowing-down of response times at boundaries compared to non-boundaries (e.g., Zwaan, 1996). Moreover, recent work has linked these changes in attention to downstream memory effects (Heusser et al., 2018), including impaired temporal order memory effects seen in our current study. Thus, we'd expect that dynamic fluctuations in attention (which, notably, are constrained by arousal states; Yerkes and Dodson, 1908) play a critical role in transforming continuous experience into discrete mnemonic events.

In addition, we'd also expect there to be variability across participants in these pupil-memory relationships, given that there is also significant variance in people's tendencies to detect event

boundaries (Zacks, Speer, Vettel, & Jacoby, 2006), and in cognitive abilities that may relate to event segmentation processes (e.g., working memory; Jafarpour et al., 2019).

Comment 2E: *The later component (2) overlaps nicely in time with previous findings, but the current approach doesn't allow a further specification of its functional significance. In addition, there is, in my view, no reason to see these effects only in experiment 3. If the pupil effect is indeed related to the formation/updating of event models, it should be present in both experiments.*

RESPONSE: We agree that component #2 (in the original submission) is a nice match with prior findings. It is indeed difficult to disentangle the behavioral/cognitive significance of this component in the current design. However, now that we have used Spearman's rank correlation approach to these analyses to mitigate the influence of outliers, we no longer see a significant correlation between this intermediate component (component #2) and temporal order memory. Thus, we have removed discussion of the intermediate component in the Discussion section and don't place much emphasis on the functional significance (main effect of boundaries) in the manuscript, as that would distract from our goal of identifying relationships between pupil measures and memory.

Comment 2F: *Similarly, if only arousal mediates event segmentation, one would have expected a relationship between the pupil correlate of arousal and the reliable source memory boost for boundary items. This suggests again that a further specification of the mediating factors of event-segmentation driven mnemonic phenomena is warranted. In summary, the correlational approach trying to link the pupil correlates of arousal to event segmentation, and later memory separation effects provided mixed results.*

RESPONSE: Reviewer 2 raises a good point. We did hypothesize that spikes in arousal at event boundaries, as indexed by pupil dilation, could also be associated with enhanced source memory binding for boundary representations (see Clewett et al., 2019 for detailed background). However, we'd like to clarify that we are not arguing that *only* arousal is involved in facilitating event segmentation. Rather, arousal is one contributing factor (or index of underlying cognitive processes) that is likely interacting with (or signaling) other complex memory and attention processes. For instance, this could include cognitive processes supported by the PFC and regions in MTL that aren't fully captured by pupil measures (e.g., Kurby and Zacks, 2010; Clewett et al., 2019). Because our previous work has consistently identified temporal memory and source memory effects at boundaries, we believe this was a sensible hypothesis. But, as the reviewer points out, we do not see evidence that pupil components relate to source binding at boundaries. We actually view this null effect as, in and of itself, quite interesting rather than problematic. It suggests that certain attentional and experimental conditions may be necessary to uncover the influence of boundary-related arousal on source binding.

This is, to our knowledge, the first empirical paper that has examined the relationship between pupillometry, arousal, event boundaries, and different forms of memory; as such, we think this

result will be meaningful and important as the field develops. If speculating, it is likely that the arousal induced by boundaries interacts with attentional processes. It may be the case, for example, that arousal will selectively enhance memory for a highly task-relevant, to-be-bound source feature (Mather and Sutherland, 2011). In our manipulation, the context manipulation is largely incidental to the main task aside from the instruction to switch hands. The hand-switching, however, has no relationship with the features of the stimuli themselves (perceptual features or semantics). Much work on arousal from the emotional memory literature suggests that arousal will only enhance binding for information that is actively attended and implicit or intrinsic to the memoranda (Mather, 2007). We believe these issues will inspire many interesting lines of research in the future. Future work will aim to combine pupil component analysis with eye-tracking as well as manipulations of attention towards different features of the memoranda.

Comment 3: The authors may want to consider a ‘subsequent memory’ approach when relating pupil responses during memory formation with later memory separation effects. For example, given an adequate number of trials available, one could use linear discriminant analysis to classify pupil responses during boundary trials as a function of later memory judgments (within participant). Interestingly, when only examining boundary marker trials, would the pupil dilation response be predictive of later judgments of timing, order, and source memory?

RESPONSE: We appreciate Reviewer 2’s excellent suggestion and for pointing out that there are additional trial-level, sensitive measures that can be leveraged to test our hypotheses. Because the paper is already quite dense and this wasn’t part of our original analyses, we report these new analyses/results briefly in the main text (pages 12-13 of Results) and in greater detail in the **Supplementary Analysis #2** (page 2 of **Supplementary Materials**; also see **Figure S3**). In short, we used linear mixed effects models (using the lmer function in R) to test whether trial-level aspects of pupil size (average pupil dilation and average pupil diameter) were related to subsequent source and temporal memory outcomes for boundary pairs.

In the first set of analyses, we modeled trial-level pupil dilation responses to the boundaries, which were computed according to the time window displayed in **Figure 5a**; namely, the average pupil size 1-2 seconds post-tone minus the average pupil size during the 500ms prior to the tone onset. In the second set of analyses, we modeled the relationship between average pupil size during the baseline period (pre-tone 500ms average) to examine how tonic fluctuations in arousal might relate to subsequent memory effects. Because the values are normalized and computed in a hierarchical linear mixed model, they reflect the tonic arousal differences within individual participants across the task. Of note, we used this latter analysis to also address Reviewer 2’s Comment #6.

In short, for the majority of analyses, we did not find any significant relationship between these phasic/tonic pupil measures and memory in any experiment (see **Figure S3**). These findings suggest that trial-level pupil-averaging might obscure more nuanced arousal/pupil mechanisms detected by the PCA. From a methodological standpoint, this also highlights the utility of using a PCA approach to dissociate different sub-components of the pupil response.

Comment 4: *It is not entirely clear in which temporal window the pupil signal was examined and baselined. From the information on pages 15, 19, and Fig 1 it appears that there is a fixation cross presented for 3 seconds before a stimulus. During this period, the onset of the tone occurs 1.5 seconds after the onset of the fixation cross and lasts for 1 second. The fixation cross is then on for another 500 ms before a picture is shown. The pupil is baselined in the period 500 ms before the tonal onset. This should thus give rise to a period of 1.5 seconds of baselined pupil data during the fixation cross. However, when looking into Figure 6b, 6c, and 6d, the analyzed baselined pupil signal comprises 3 seconds. It thus seems as if the last 1.5 seconds of this time window then represents 1.5 seconds during the succeeding picture presentation. I may, of course, have misunderstood the trial structure, but if the interpretation is correct, this means that the first and second half of the pupil data belong to two different visual inputs, which may severely affect the data and PCA. Please clarify.*

RESPONSE: ****Please note that the figure numbers Reviewer 2 is referring to have now changed. These figures are now numbered Figure 5a, and 5d in the revised manuscript.**

We apologize for any confusion about the time-window. Reviewer 2 is correct: the pupil-sampling window was 3s long and started at the onset of the tone. This means it captured the 1.5 seconds of fixation starting at the onset of the tone as well as the first 1.5 seconds of the subsequent image.

Having clarified this, however, we agree with Reviewer 2's suggestion that examining the shorter window (1.5s) would provide additional information about how tones are affecting pupil dilation independently of the memoranda. We now report a shorter time-window (1.5s) for the main PCA in the **Supplementary Analysis #1** (pages 1-2 of **Supplementary Materials** and **Figure S2**). Below, we briefly describe our logic for focusing on the original 3s PCA in this manuscript.

Logic for conducting a 3s PCA window

In the original manuscript, we chose the 3s window *a priori*, because – as Reviewer 2 and 3 point out – we were interested in whether arousal responses predict enhanced source binding at boundaries. Much research from our lab and others (see Clewett et al., 2019 for review; Rouhani et al., 2019; Siefke et al., 2019) demonstrates that source memory is typically enhanced for boundary representations; that is, for information that appears at the boundary, or context shift. Further, in the attention domain, there is often a slowing in reaction time judgements for boundary items versus within-event items (e.g., Heusser et al., 2018), which is often interpreted as increased attention to and/or prioritization of boundary information. Based on those findings, we predicted that the “boundary-ness” of both the context shift and the first item in an event may relate to a spike in arousal/pupil dilation (captured by pupil dilations to both the tone and its following item), given that boundary information also appears to be processed differently in attention and memory than within-event, or same-context, information.

With respect to Reviewer 2's concern about the first and second half of the pupil response belonging to “two different visual inputs”, we don't view this as problematic, because this would,

from a methodological perspective, be equally confounding for all trial types: each trial and condition would be affected by this carryover to the same degree. In essence, any pupil contamination is inherently controlled for in our comparisons. We again emphasize that our pupil component-memory correlations were also derived from subtraction scores. Furthermore, we reiterate that the luminance of all visual stimuli (both the fixation and images) was matched. As such, there is not a systematic bias between the effects of the images and the tones on the pupil light reflex (i.e., light and sensory-driven effects that may confound pupil indices of cognitive processing).

Logic for including a new 1.5s PCA window in the revised supplementary material

To address Reviewer 2's helpful suggestion, we now report a 1.5s PCA analysis that focuses on a smaller time window (1.5s). This enabled us to specifically target the effects of the tones (the fixation period 1.5 seconds after tone onset, or 375 pupil samples).

Importantly, this new, 1.5s PCA revealed three of the same canonical components identified in the original, longer time-window PCA reported in our original paper (see **Figure S2**). This adds confidence that our pupil PCA measures reflect stable underlying mechanisms that regulate pupil size and are engaged by task-relevant tone switches in this task. When examining this specific tone-related time window, we also found interesting main effects of boundaries on the different pupil loadings. Specifically, compared to the pupil analysis from the 3s PCA (see **Figure 5d**), we now see changes in two of the components' loadings at boundaries. For convenience, this figure is also displayed below:

Supplementary Figure 2. (a) Temporal features of tone-evoked pupil dilation identified by a temporal principal component analysis (PCA) focused on the 1.5s tone period. For illustrative purposes, separate PCAs were performed on the boundary and same-context trials to show qualitative differences in pupil loadings across time. Vertical dashed lines signify the onsets of the tones and their subsequent images. The 1.5s PCA revealed three significant features of pupil dilation that had distinct shapes over time and matched three of the components identified in the 3-s window PCA. Component loadings reflect “raw” values from the rotated solution, so are on the same scale as the original pupil inputs. (b) Condition-related differences on pupil component loadings. The boundary tones (blue colors) versus same-context tones (gray) differentially modulated loading scores for component #1 (intermediate) and component #3 (early-peaking), but not for component #2 (slowly-decreasing). All error bars represent standard error of the mean (SEM). *** $p < .001$; ** $p < .01$; ~ $p < .10$.

First, we no longer see a significant boundary effect on the ‘anticipatory’ pupil component (#3 in the main PCA and now #2 in the 1.5s PCA; compare **Figure 5d** in main text and **Figure S2b** above), suggesting that “boundary-ness” (tone + item) matters. We interpret this pupil response as reflecting that participants actively anticipate not only the tone changes, but also their responses to the impending boundary image. In effect, this dilation is reflecting the emergence of a new ‘event model’ related to changes in top-down cognitive processing/representations of

the new event (e.g., Zacks et al., 2007). We would not necessarily expect to see the emergence of this event model before participants have begun to process stimuli within the new event. In other words, the combination of the contextual information and the first item embedded within it both appear to be important for how memories become organized into events later on.

Second, in the short 1.5s PCA, we now see a significant effect of boundaries on loading onto the “early-peaking” component (#4 in the 3s PCA and now #3 in the 1.5s PCA; **compare Figure 5c and Figure S2**). As shown in **Figure 6c**, there is a clear qualitative change in when individuals load onto this component. While this component (#4 in the long, 3-s window analysis) is evident during the boundary tones, same-context images, and boundary images, it is not present during the same-context tones **Figure 6c**. This is important, because the same-context tones were the only stimulus/timepoint that did not require a motor change or task-relevant response.

In conjunction with prior work linking pupil peaks at this specific timepoint to motor processes (Beatty, 1982; Richer et al., 1983; Shiga and Ohkubo, 1979; Steinhauer and Hakerem, 1992), this again reinforces the idea that participants are reconstructing a new event model (or cognitive representation/motor-mapping) at boundaries in ways that promote the segmentation of those events in memory (given that loading on this component was related to subjective changes in temporal distance ratings later on). This boundary-related loading effect on the “early-peaking” pupil component was not observed in the 3s PCA (see **Figure 5d**), suggesting that loading differences at boundaries were obscured by the strong motor-related response to the same-context images (see rightmost red arrow/peak in **Figure 6c**).

Comment 5: *I also think it would be valuable to look into pupil size in the baseline period. A growing literature within the adaptive gain theory, looking at exploration and exploitation control states of behavior has demonstrated that an exploratory mode is characterized by larger baseline pupil size (tonic changes) and an exploitative mode by phasic changes during the task period (e.g., Jepma et al., 2011). I think this could be quite revealing when examining the mechanisms underlying pupillary changes around event boundaries. Why not use the initial fixation cross in each trial for baseline correction? In this way, one would be able to track any anticipatory pupil response and disentangle this from tone-evoked responses.*

RESPONSE: This is an excellent suggestion. We now report the results of these linear mixed effects models in the **Supplementary Analysis #2** on page 2 of the **Supplementary Materials**. These new methods/results are described in more detail in our response to Reviewer 2’s Comment 3 above. Briefly, we did not find any significant effects of tonic pupil size during the baseline period on temporal distance memory or temporal memory in either experiment. We mention this null result in the Results section on pages 12-13 of the main text.

Comment 6: *The units for pupil diameter should be pixels or mm (Figures 6a and 6b).*

RESPONSE: ****Please note that the figure numbers Reviewer 2 is referring to have now changed. These figures are now numbered Figure 5a and 5b, respectively.**

As specified in the Eyelink 1000 manual, the eye-tracker records pupil diameter in arbitrary units:

“The pupil size data is not calibrated, and the units of pupil measurements will vary with subject set-up. Pupil size is an integer number, in arbitrary units” (page 100).

Based on this specification, we have not changed the units to pixels or mm. We also note that it is often the case in the literature that pupil units aren’t specified, as these studies also likely record pupil diameter in arbitrary units from the eye-tracker.

Comment 7: *Figure 6d is a bit confusing. It would be preferable to see the real factor scores and not the normalized scores (i.e., relative difference centered around 0). Otherwise, it is difficult to interpret the magnitude and direction of those effects.*

RESPONSE: ****Please note that the figure number Reviewer 2 is referring to (6d) has now changed. In the revised version this is Figure 5d.**

We thank Reviewer 2 for pointing this out. Unfortunately, however, we are not entirely clear what is meant by “real” factor scores.

Perhaps Reviewer 2 is referring to “factor-based” or “item-based” scores, which tend to provide a more intuitive measure since it places the component loadings on the same scale as the input data? This approach is somewhat challenging with a temporal PCA approach, as there are 750 dependent measures that are based on specific timepoints/samples. Our understanding is this approach would be helpful for more intuitive measures, such as items on a questionnaire. In that case, the items (DV’s) can be assigned to their latent factor and averaged, placing them on the same scale. For instance, different questions on an anxiety questionnaire can be assigned to the factor they load onto most and then averaged together. But this strikes us as being impractical when using variables that reflect discrete timepoints as opposed to behavioral scores or performance measures.

Even so, we are able to display the “raw” loadings for each component across time and across the *entire group of participants*. We now display the raw PCA component time courses and the pupil loadings to illustrate the direction and relative magnitude of the effects in **Figure 5c**, **Figure 6**, and **Figure S2a**. Importantly, however, we cannot extract individual “raw” factor scores for each participant, which would enable statistical comparisons between conditions. These plots simply represent the loadings of each component across time and across all of the pupil data input into to the PCA.

We would also like to note that, in the literature on pupil temporal PCAs, it is often standard to report the normalized factor scores and perform statistical comparisons on those values (e.g., Johansson et al., 2018). Furthermore, one advantage to PCA is it measures how “representative” these temporal patterns are within an individual (i.e., the loadings) rather than

the actual magnitude (size) of the sub-components of pupil dilation. In other words, the component loadings aren't driven by an individual showing a larger peak pupil dilation, or change in overall pupil size. In this sense, we don't view the absolute magnitude as being informative above and beyond what it is qualitatively shown in the time-course figures.

Reviewer #3 (Remarks to the Author):

Summary: This manuscript reports three experiments exploring how a surprising change in context or event boundary elicits pupil dilation, as well as changing in memory for perceptual qualities such as timing and order. Importantly, different pupillary response components were associated with different cognitive components. For Experiment 1, people remembered objects after an event boundary worse than those not at an event boundary. Moreover, responses were consistent with the idea that event boundaries increased the perceived time duration across them, and relative temporal order memory was worse. Finally, memory for the ear of a tone heard at a boundary was worse than when the objects were not at an event boundary. Experiment 2 replicated Experiment 1 with the addition of pupillometry measurements. The basic results of Experiment 1 were replicated. Experiment 3 replicated Experiment 2 with the addition of a change in the actual temporal distance between items. The basic results of Experiments 1 and 2 were replicated. Moreover, in Experiments 2 and 3, the pupillometry data revealed increased pupil size at event boundaries, consistent with the idea that event boundaries are accompanied by increased arousal. The results are interpreted as showing that there is physiological arousal at event boundaries.

Recommendation: This is a well-written manuscript that explores interesting some ideas. However, I do have some points of concern that are listed below.

Major Points

Comment 1: *The behavioral results reported here state that memory is worse for objects around or at an event boundary. However, other research shows the opposite (e.g., Swallow, Zacks, & Abrams, 2009). How can this be accounted for? Is it just the case that the tone switch serves as a distractor taking attention away from the main task?*

RESPONSE: We apologize for any confusion. It is true that Event Segmentation Theory (Zacks et al., 2007) postulates, and some empirical work has shown, that information encountered at event boundaries is later better recognized than items presented within events (Swallow et al., 2009). We also see that source memory (memory for the object-ear pairing, in this case) is indeed better for boundary items compared to same-context items. This replicates prior work. However, several studies from our lab and other labs have now shown that temporal memory across boundaries is significantly impaired and is better within events. The memory *impairment* that we report concerns temporal order memory, which is consistent with prior event boundary studies (e.g., Clewett and Davachi, 2018; Davachi and Dubrow, 2013; 2014; Heusser et al., 2018).

Comment 2: *The pupillometry results reported here suggest that memory is worse for objects around an event boundaries when arousal is greater. However, other research shows that increased arousal leads to better memory (e.g., LaBar & Phelps, 1998). How can this be accounted for?*

RESPONSE: We again apologize if we created any confusion about our memory impairment results. As we clarified in the previous response, we did not test object recognition in the current study (as many emotional arousal studies do). Prior work has shown that arousal can increase memory for the arousing items. Here we are measuring temporal integration mechanisms (inter-item relationships) and how they are influenced by arousal at event boundaries. Consistent with earlier work, we show that manipulating boundaries event boundaries consistently leads to a decrease in temporal integration, which was measured as worse temporal order memory and larger temporal distance estimates for boundary versus same-context item pairs.

With respect to source memory for the information presented at the boundary, it was indeed possible that arousal at boundaries may lead to increased source binding for the boundary representations. And that was one of our predictions at the beginning of this study. The situation, however, is complicated by the fact that arousal acts as a double-edged sword that can either *enhance* or *impair* episodic memory (see Mather and Sutherland, 2011 for a review). While it may be surprising that arousal didn't relate to enhanced source memory binding in our studies, there are many factors that dictate whether arousal will enhance or impair source binding. Indeed, the findings in the emotional arousal literature are often mixed, particularly with respect to source memory. These factors include – but are not limited to – the timing between the arousal and the memoranda and the priority of the boundary source information (e.g., relevance to encoding).

The main factor that we think is critical and could explain why we do not see source memory increases with arousal is that our event boundaries were separated in time from the presentation of the memoranda. The transient nature of phasic arousal effects on memory has been demonstrated in many studies of emotion and memory, with the temporal distance between the arousing stimulus and a subsequent neutral item dictating whether arousal will enhance, impair, or have no effect on memory for that item (Bocanegra and Zeelenberg, 2009). Thus, the time-course of arousal is quite important for determining the fate of nearby information in memory. In our current design, the memoranda were presented 1.5 seconds after the tone, which may have been too long of a time-window to uncover any modulation of source memory by arousal. Future could manipulate the length of this window to see whether a potential relationship between boundary-related arousal and source memory is time-dependent (e.g., Swallow and Jiang, 2014; see also Clewett et al., 2019 for more details).

Comment 3: *Do the pupillometry data reflect event boundaries, or surprise at the tone shift? There is some evidence that surprise and event boundaries, while they may co-occur, do not always. In those cases, effects of the event boundaries and surprise can be separated out*

(Pettijohn & Radvansky, 2016). Given that it is difficult to do so with this data, can anything be concluded about the cause of the observed patterns of performance?

RESPONSE: This issue of surprise at boundaries being involved in these effects is a valid point. We'd argue, however, that it's difficult to view these effects as being driven by 'surprise', per se, because the event structure in the task was predictable; that is, the tone would always switch to the other ear after 8 successive items. Surprise is typically defined as events/stimuli that occur at random, or involve 'unexpected uncertainty' (Yu and Dayan, 2005). Further, the sensory features of the stimuli within the experiment are equivalent across conditions. By subtracting the pupil component loadings between the boundary and same-context tones in our correlation analyses, we were able to isolate the key difference between them: the task relevance, or motivational significance, of the boundary switches.

Reviewers' comments:

Reviewer #2 (Remarks to the Author):

R1 "Dynamic arousal signals relate to memories of time and events"

Clewett, Gasser & Davachi

As evident in my original review, I think this study addresses an important and timely research question that should be of interest to a broad readership. My main concerns pertained to whether the theoretical contribution of the manuscript is sufficient to warrant publication in Nature Communication. The authors are to be commended for conducting an extensive revision of their paper, including a better anchoring of the study in the existing literature, and having added several informative analyses and re-analyses of their data. Together, these changes have improved the manuscript considerably. Despite this, however, I remain unconvinced that these data offer conclusive evidence concerning the role of arousal in event segmentation and in mediating later temporal memory effects. Please find below some remaining comments that the authors may want to consider in any further revision of their manuscript.

The major novel claims of the paper are supported by correlations between pupil changes evoked by context changes (auditory tone switches left/right ear) and later memory phenomena driven by the same context changes (i.e., boundary condition). While a vast amount of previous work has demonstrated both of these consequences of an event boundary, the current study extends this by showing that dissociable components of the pupil data co-vary with inter-individual variability in later temporal memory judgements. Although potentially very interesting, these findings cannot be interpreted in terms of causality; thus, claims that arousal mediates/forms/shapes/affects/modulates/etc. need rephrasing.

One problem is that the boundary tones are both salient and task-relevant, which complicates the interpretation of the functional significance of the pupil changes. Are they driven by stimulus salience or by task relevance? The authors claim they are "effectively controlling for lower-level sensory properties of the stimuli" since the only difference between boundary and same-context tones is the channel change and the re-mapping of motor demands. They consequently favor interpretation in terms of task relevance. However, although the sensory features of the tone may be comparable for boundary and same-context tones, only boundary tones involve a switch from one ear to another, a change that should be salient with or without the need to update behavior. The paradigm would have benefitted from a baseline condition involving task-irrelevant auditory stimuli changes, to allow firmer conclusions about the functional significance of the pupil changes.

The new PCAs conducted separately for boundary and same-context conditions are interesting (Fig 6). The early-peaking component (#4) is only present for boundary trials and the authors interpret this in terms of the abovementioned re-mapping of motor responses. I have a couple of questions related to this:

- Did you perform a statistical test to validate the difference between boundary and same-context (e.g. presence of an early peaking component on boundary trials only)?
- Do component #4 factor loadings (Boundary 3s Pupil PCA) predict inter-individual differences in the temporal memory effects observed in Exp 2 and 3? Such an analysis may provide a more sensitive assessment of the relationship between boundary-induced pupil responses and later subjective temporal memory effects?
- Is motor re-mapping the only viable interpretation of pupil changes in the boundary condition? Differences in motor demands are not the only difference, as only boundary trials involve a spatial switch of the tone presentation. Interestingly, the tone is irrelevant until it changes, and changes should trigger re-allocation of attention. Thus, an alternative to the re-mapping of a task/motor response interpretation would be to interpret the effect as attention allocation to task-relevant auditory changes. Such re-allocation of attention could be initiated by a prediction error (driven by tone update). The authors may want to consider discussing alternative interpretations of the functional significance of this PCA component. Given the event-segmentation framework, signals of prediction error and attention re-allocation may be relevant alternatives.

The authors' "goal was to identify pupil-memory associations that could account for variability in boundary-related arousal and memory difference across participants". Indeed, the main novel finding of the study was correlations between the pupil and temporal memory effects (selectively for components #3 and #4, explaining 3.01% and 2.66% of the total pupil variance, respectively). As pointed out in my original review (Comment 2B), the "early peaking" component #4 was not sensitive to the main manipulation used to implement the boundary/same context logic in the experimental paradigm (Fig 5d Comp. 4). Nevertheless, the authors decided to correlate this component with temporal memory effects considered to be driven by such boundary/same context changes. If there is no reliable pupil difference between boundary and same-context trials it seems incorrect to refer to "boundary-related arousal" effects. In their rebuttal letter, the authors state that "some pupil components reflect more generalized cognitive processes that facilitate information processing as events unfold". While this certainly is feasible, it doesn't seem to match the interpretation that the authors favor regarding the functional significance of the component, namely that it corresponds to re-mapping of motor demands due to the boundary trials. Thus, I believe that the relevance of my previous comments remains. Moreover, regarding the authors continued line of reasoning, I do not immediately see how a subsequent memory analysis using fMRI would be at all analogous to what the authors are doing here (i.e., correlating a pupil null effect with a reliable behavioral effect). What exactly does "in the absence of a main effect of memory performance in behavior" mean? An analogy to a subsequent memory analysis would be a lack of a BOLD response difference at encoding between subsequently remembered and forgotten items. Or, perhaps that the subsequent memory effects for deep vs shallow encoding are identical. It would seem inappropriate to then use the non-existing BOLD difference between deep and shallow to try to predict the mnemonic benefit for deeply encoded items. Perhaps the authors mean that subsequent memory effects could be participant-specific and thus failing to hold in a group analysis. Nonetheless, such a pattern would have been the result of an analytical approach involving statistical tests within participant, and no such statistics were conducted here.

The authors have added a PCA of the pupil data restricted to the first 1.5s time window of each encoding trial as suggested in the previous review. This analysis should offer an even better insight into the tone-induced boundary effects as it effectively eliminates the impact of pupil changes driven by the onset of the image. The “early peaking” component remains and is now significantly different for boundary and same-context trials. Why are these data not used to predict subjective temporal memory?

More generally, it is unclear why the authors' primary focus is on arousal rather than on particular mechanisms, such as motor re-mapping, prediction error signaling, reorientation of attention, updating, etc., that are affected by contextual changes and presumably involved in event segmentation. The PCA approach adopted here to examine the pupil data seems well-suited for such purposes, yet the authors make inferences in terms of the broader concept of arousal. It is not apparent why arousal dynamics need to be interpreted as the cause of event segmentation and later memory phenomena, rather than being sensitive to and interacting with mechanisms such as the ones mentioned above. The arousal signals could, of course, be the cause, but then the experimental approach would need revision to be able to establish that. Currently, it seems more appropriate to conclude that the pupil tracks information processing influenced by event boundaries and that dissociable pupil components may reveal the contribution of distinct mechanisms with potentially different relevance for later subjective temporal memory. PCA decomposition seems promising to establish further which these mechanisms are and to specify their relative contributions.

In short, I find this paper very interesting, reporting several findings that may indeed prove important. Nevertheless, I also note several weaknesses that hamper the interpretation of the functional significance of the observed pupil-memory association. Thus, I remain uncertain as to whether the current package makes a sufficiently strong contribution to warrant publication in Nature Communication.

Reviewer #3 (Remarks to the Author):

MS. NCOMMS-19-32362

Title: Dynamic arousal signals construct memories of time and events

Authors: Clewett, Gasser, & Davachi

Summary

This is a revised version of a manuscript that I had reviewed earlier.

Recommendation

This is an improved version of the prior manuscript, and I am satisfied with the responses to the reviews.

Response to Reviewer 2's Comments

Summary: *As evident in my original review, I think this study addresses an important and timely research question that should be of interest to a broad readership. My main concerns pertained to whether the theoretical contribution of the manuscript is sufficient to warrant publication in Nature Communication. The authors are to be commended for conducting an extensive revision of their paper, including a better anchoring of the study in the existing literature, and having added several informative analyses and re-analyses of their data. Together, these changes have improved the manuscript considerably. Despite this, however, I remain unconvinced that these data offer conclusive evidence concerning the role of arousal in event segmentation and in mediating later temporal memory effects. Please find below some remaining comments that the authors may want to consider in any further revision of their manuscript.*

Comment #1: *The major novel claims of the paper are supported by correlations between pupil changes evoked by context changes (auditory tone switches left/right ear) and later memory phenomena driven by the same context changes (i.e., boundary condition). While a vast amount of previous work has demonstrated both of these consequences of an event boundary, the current study extends this by showing that dissociable components of the pupil data co-vary with inter-individual variability in later temporal memory judgements. Although potentially very interesting, these findings cannot be interpreted in terms of causality; thus, claims that arousal mediates/forms/shapes/affects/modulates/etc. need rephrasing.*

RESPONSE: We agree that we cannot make inferences about arousal having a *causal* influence on temporal memory effects. While it is certainly possible that arousal-mediated cognitive (e.g., prediction errors, attention re-orienting, cognitive effort, etc.) and neuromodulatory (e.g., NE, DA, Ach) processes may play a direct role in modulating event segmentation in memory, we cannot rule out the non-mutually exclusive possibilities that these arousal signals are simply sensitive to and/or interacting with other brain mechanisms that support attention and episodic memory organization. Considering this, we have included this type of language (e.g., “tracks”, “is sensitive to”, “signals”, “relates to” etc.) throughout the manuscript.

The strong advantage of PCA decomposition is that different aspects of the pupil response may be tracking different types of cognitive processes (and even autonomic nervous system processes; e.g., Steinhauer and Hakerem, 1992) that modulate objective and subjective aspects of temporal memory. We have now taken a more agnostic/objective stance on what pupil-related arousal may signify in this paradigm, and have re-phrased our description of arousal responses throughout the manuscript and in the new manuscript title. Wherever it is relevant, we also refer to the actual physiological measure (pupil) and describe arousal processes as “pupil-linked arousal”.

Comment 2: *One problem is that the boundary tones are both salient and task-relevant, which complicates the interpretation of the functional significance of the pupil changes. Are they driven*

by stimulus salience or by task relevance? The authors claim they are “effectively controlling for lower-level sensory properties of the stimuli” since the only difference between boundary and same-context tones is the channel change and the re-mapping of motor demands. They consequently favor interpretation in terms of task relevance. However, although the sensory features of the tone may be comparable for boundary and same-context tones, only boundary tones involve a switch from one ear to another, a change that should be salient with or without the need to update behavior. The paradigm would have benefitted from a baseline condition involving task-irrelevant auditory stimuli changes, to allow firmer conclusions about the functional significance of the pupil changes.

RESPONSE: We agree that including a task-irrelevant tone baseline condition would aid our interpretations in general. However, we don't believe it is necessary for drawing conclusions about the functional significance of pupillary component #4. If component #4 loadings were primarily driven by salience, then we would not expect it to appear in response to same-context images (which are presumably less interesting and salient than the other tone and image types). However, we see evidence of component #4 loadings for same-context stimuli in our paradigm, suggesting that salience is not the driving force behind this pupil component (see **Figure 6c**).

Furthermore, while not explicitly linked to the PCA-memory correlations, the results of our linear mixed modeling analyses (**Supplementary Analysis/Results #3; Figure S4a**) revealed no significant relationships between trial-level pupil dilation and the amount of change (pitch change) across boundaries.

Based on existing PCA pupil studies, we'd also expect pupil component #2 to be a potentially better indicator of stimulus salience, given that this component is modulated by salient (e.g., pink noise, phone ringing) and highly emotional auditory stimuli (Wetzel et al., 2016; Widmann et al., 2018). But interestingly, boundary-related loadings on this pupillary component (#2; intermediate) were not significantly correlated with boundary-related memory outcomes in our study. In sum, we think these points provide an argument that it is unlikely that the component #4 correlation with temporal memory (and our pupil measures more broadly) specifically relate to any effects of the salience of event boundaries.

On a more practical level, we did not include a baseline tone switch manipulation in this experiment, because we were and remain primarily interested in testing how and whether pupillary components relate to temporal (and source) memory behavior. The addition of a baseline auditory tone that did not signify any sort of task change may or may not have produced event segmentation in the moment or in subsequent memory; thus, we would not have known whether we had sufficient power in our design to properly assess memory behavior. Nevertheless, including within-subject manipulations of the task relevance of the tone switches would be an interesting direction for future research and we thank the reviewer for his/her thoughtful comments.

Comment 3: *The new PCAs conducted separately for boundary and same-context conditions are interesting (Fig 6). The early-peaking component (#4) is only present for boundary trials and*

the authors interpret this in terms of the abovementioned re-mapping of motor responses. I have a couple of questions related to this:

RESPONSE: For clarification, when Reviewer 2 states that “*The early-peaking component (#4) is only present for boundary trials,*” he/she is likely referring to only the tone-induced pupil response (see **Figure 4c**; red-shaded box is where component #4 is not evident for same-context tones). Component #4 is present in both conditions (boundaries and same-context trials) when both the image and tone periods are combined/considered together.

Comment 3A: *Did you perform a statistical test to validate the difference between boundary and same-context (e.g. presence of an early peaking component on boundary trials only)?*

RESPONSE: Yes, this statistical comparison was done using the 1.5s PCA, which isolates the only stimulus timepoint when component #4 was not evident (i.e., the same-boundary tone; for visual comparison, see **Figure 6c and Figure S2**).

As shown in **Figure S2b** (turquoise bars) and reported at the bottom of page 1 of the **Supplementary Materials**, boundary tones led to significantly greater loadings on pupil component #4 compared with same-context tones (turquoise bars; $t(64) = 3.51$, $p < .001$ [CI: 0.21, 0.76]).

Comment 3B: *Do component #4 factor loadings (Boundary 3s Pupil PCA) predict inter-individual differences in the temporal memory effects observed in Exp 2 and 3? Such an analysis may provide a more sensitive assessment of the relationship between boundary-induced pupil responses and later subjective temporal memory effects?*

RESPONSE: We believe that our subtraction approach is optimal for testing our main hypotheses, because subtracting out the pupil effects on same-context trials helps control for sources of within-subject variability prior to the performing across-subject correlations. More importantly, performing subtractions is the most appropriate way of testing for boundary-specific effects on cognition and memory. This enables us to test our main prediction that different pupil components at boundaries may track and/or contribute to subsequent event segmentation effects in temporal memory.

Consistent with our key hypotheses, these subtraction/correlation analyses revealed that different pupil-linked components of arousal were associated with both subjective and objective aspects of temporal memory. Specifically, we found that greater boundary-related factor loadings on pupil component #4 were associated with larger retrospective estimates of temporal distance between pairs of items that spanned a boundary (tone switch) compared to items pairs encountered within the same auditory context (repeated tone). We also found that greater boundary-related factor loadings on pupil component #3 were associated with worse temporal order memory discriminations for item pairs that had spanned boundaries compared to item pairs that had been encountered within the same context. Together these results suggest that

boundary modulations of pupil PCA components scale with the magnitude of boundary modulations of temporal memory across individuals.

As described in the previous paragraph, we don't believe only analyzing the boundary trials is the best way to test our specific hypotheses study and to draw interpretations about how pupil-linked arousal effects at boundaries signal/contribute to changes in episodic memory. For the sake of completeness, however, we have run these analyses to directly address this specific question from Reviewer 2. When we examined the relationship between pupil component #4 factor loadings and individual differences in temporal memory for boundary trials only (3s PCA), we found that: Across experiments, boundary factor loadings on component #4 showed a statistical trend towards a significant correlation with temporal distance ratings ($\rho = .20$, $p = .11$). This effect was also a statistical trend in Experiment 3 ($\rho = .26$, $p = .16$) and was not significant in Experiment 2 ($\rho = .12$, $p = .49$).

Comment 3C: *Is motor re-mapping the only viable interpretation of pupil changes in the boundary condition? Differences in motor demands are not the only difference, as only boundary trials involve a spatial switch of the tone presentation. Interestingly, the tone is irrelevant until it changes, and changes should trigger re-allocation of attention. Thus, an alternative to the re-mapping of a task/motor response interpretation would be to interpret the effect as attention allocation to task-relevant auditory changes. Such re-allocation of attention could be initiated by a prediction error (driven by tone update). The authors may want to consider discussing alternative interpretations of the functional significance of this PCA component. Given the event-segmentation framework, signals of prediction error and attention re-allocation may be relevant alternatives.*

RESPONSE: We agree with Reviewer 2 that there may be different (yet non-mutually exclusive) ways to interpret the functional significance of pupil component #4. However, we do not think prediction error is the most feasible explanation. We are aware that prediction error has appeared in theoretical work to be related to boundaries (e.g., Reynolds et al., 2007; Zacks et al., 2007) but little empirical evidence has modulated prediction error (see also Clewett and Davachi, 2017 for additional discussion on these limitations). We are sensitive to the fact that prediction error can mean different things in different fields and is widely used in perception, reward and now the memory literatures. As such, we also believe that it may be underspecified as a construct, especially within the context of episodic memory. In our design, the presence of the boundary tone switch was completely predictable, as it happened every 9th trial and every 44 seconds.

From an empirical perspective, we might also expect a prediction error response to be signaled by loading on pupillary component #2 (intermediate component) rather than pupillary components 3 or 4 (the two components that were related to memory behavior in our study). This hypothesis is based on evidence that average pupil dilation responses to prediction errors have been shown to peak within a window consistent with the peak timepoint of component #2 loadings (~1.5-2 seconds post-error; Alamia et al., 2019). We did not find any significant correlations between boundary-related loadings on component #2 and memory, lending

additional indirect support for the idea that it is unlikely that our pupil results represent a prediction error mechanism of event segmentation.

Having said that, we do agree that some versions of a prediction error account (e.g., ones that assume the brain is only sensitive the current state and any change from that state may represent an error) may account for boundary-related changes in pupil dilation and memory segmentation more generally. We now include the following text where we previously discussed the potential influence of predictability on event segmentation in memory:

“One interesting possibility is that unpredictable event boundaries also elicit event segmentation in memory by triggering arousal-related prediction error signals that facilitate event-model updating (e.g., [64]). However, this model-updating process may be signaled by other temporal components of pupil dilation.”

Comment 4A: *The authors’ “goal was to identify pupil-memory associations that could account for variability in boundary-related arousal and memory difference across participants”. Indeed, the main novel finding of the study was correlations between the pupil and temporal memory effects (selectively for components #3 and #4, explaining 3.01% and 2.66% of the total pupil variance, respectively). As pointed out in my original review (Comment 2B), the “early peaking” component #4 was not sensitive to the main manipulation used to implement the boundary/same context logic in the experimental paradigm (Fig 5d Comp. 4). Nevertheless, the authors decided to correlate this component with temporal memory effects considered to be driven by such boundary/same context changes. If there is no reliable pupil difference between boundary and same-context trials it seems incorrect to refer to “boundary-related arousal” effects. In their rebuttal letter, the authors state that “some pupil components reflect more generalized cognitive processes that facilitate information processing as events unfold”. While this certainly is feasible, it doesn’t seem to match the interpretation that the authors favor regarding the functional significance of the component, namely that it corresponds to re-mapping of motor demands due to the boundary trials. Thus, I believe that the relevance of my previous comments remains.*

RESPONSE: We thank Reviewer 2 for making this excellent point and we apologize for any confusion regarding our interpretation of “motor re-mapping.” Although there was no main effect of the manipulation on component #4 factor loadings, we still believe that it is valid to interpret a subtraction between conditions as being indicative of “*boundary-related* arousal.” This is because the purpose of the subtraction is to dissociate *specific effects* of boundaries on pupil component loadings and behavior.

As shown in **Figure 6C**, factor loadings on pupil component #4 are only evident at the three stimulus timepoints when a motor response is required (i.e., boundary tone, boundary image, same-context image). It is not evident for same-context tones when no motor response is required. This pattern suggests that this pupil component specifically relates to any motor response, meaning it may indeed reflect a mental process that is “generalized” (i.e., motor) and not specific to task relevance or context shifts.

When we refer to motor *RE*-mapping, we are specifically referring to the subtraction measures of boundary minus same-context component #4 loadings. In essence, subtracting out the baseline effects of any motor responding on the same-context trials from loadings on boundary trials enables us to isolate aspects of the motor-related pupil dilation specifically driven by task-relevant motor *RE*-mapping. Because the primary difference between conditions is that there is a motor *change* at boundaries, we interpret this effect as signifying a *re*-mapping, or updating, effect.

Comment 4B: *Moreover, regarding the authors continued line of reasoning, I do not immediately see how a subsequent memory analysis using fMRI would be at all analogous to what the authors are doing here (i.e., correlating a pupil null effect with a reliable behavioral effect). What exactly does “in the absence of a main effect of memory performance in behavior” mean? An analogy to a subsequent memory analysis would be a lack of a BOLD response difference at encoding between subsequently remembered and forgotten items. Or, perhaps that the subsequent memory effects for deep vs shallow encoding are identical. It would seem inappropriate to then use the non-existing BOLD difference between deep and shallow to try to predict the mnemonic benefit for deeply encoded items. Perhaps the authors mean that subsequent memory effects could be participant-specific and thus failing to hold in a group analysis. Nonetheless, such a pattern would have been the result of an analytical approach involving statistical tests within participant, and no such statistics were conducted here.*

RESPONSE: We apologize if this fMRI analogy created more confusion. Let's move beyond this example to focus on the specific procedures of the current study. We'd first like to clarify that these analyses were still within participant: the pupil component factor loadings and behavioral memory measures were first subtracted within each participant prior to correlating these measures across the group. We then examined statistically how differential pupil loadings to boundary trials (boundary – same-context item pairs) were related to the differential memory measures for boundary compared to same-context trials across participants.

We underscore that this approach of performing neurophysiology-behavior linear correlations without a main effect of a manipulation are present consistently throughout the neuroscience literature. For example, one study examined the relationship between condition-specific effects (e.g., effects of direct electrical stimulation on neural drift rate in the MTL) and across-participant differences in memory, even though condition-specific effects of stimulation did not significantly differ on average (El-Kalliny, M. M., Wittig, J. H., Sheehan, T. C., Sreekumar, V., Inati, S. K., & Zaghloul, K. A. (2019). Changing temporal context in human temporal lobe promotes memory of distinct episodes. *Nature Communications*, 10(1), 1-10). Likewise, in a separate study, linear correlations were performed for corresponding subtraction scores of neurophysiological effects (i.e., hippocampal fMRI BOLD signal suppression) and memory (i.e., amnesic shadow effect) across participants, despite no significant difference, on average, in the neurophysiological effect (Hulbert, J. C., Henson, R. N., & Anderson, M. C. (2016). Inducing amnesia through systemic suppression. *Nature Communications*, 7, 11003).

In summary, the goal of our pupil-memory individual differences analyses was to target the magnitude of these effects. We were primarily interested in whether those individuals who showed the largest modulation of pupil dilation components by boundaries also exhibited the largest event segmentation effects in temporal memory. Indeed, we identified several pupil-memory correlations that are consistent with this prediction, suggesting that certain temporal aspects of the pupil-linked arousal response to boundaries track the formation of new episodes in episodic memory. Moreover, the correlation results reveal that different pupil components – which may reflect different cognitive processes (e.g., motor responses or top-down predictions) and potentially autonomic processes – contribute to boundary-related changes in objective and subjective aspects of temporal memory.

To further alleviate Reviewer 2's concern, we note that the pattern of results was the same when we performed a follow-up analysis limiting the analysis time-window (1.5s PCA) to the tone period when boundaries did significantly modulate component #4 loadings (see response to next comment, Reviewer 2 Comment #5).

Comment 5: *The authors have added a PCA of the pupil data restricted to the first 1.5s time window of each encoding trial as suggested in the previous review. This analysis should offer an even better insight into the tone-induced boundary effects as it effectively eliminates the impact of pupil changes driven by the onset of the image. The “early peaking” component remains and is now significantly different for boundary and same-context trials. Why are these data not used to predict subjective temporal memory?*

RESPONSE: It is indeed possible that focusing on the 1.5s time-window PCA would help us dissociate the specific influence of the context shifts (i.e., tone switches) on subsequent memory. However, as we described in the previous response letter, our *a priori* hypothesis was that the cognitive processes triggered by context shifts, or event boundaries, influence the processing of boundary representations (in this case, the first object image following a tone switch). Thus, before beginning this experiment, we chose to examine a broader time-window (both the tone switch and the following image; 3s PCA) of pupil samples to assess how event boundaries influence the temporal organization of events in memory.

As described in the main text on page 10, our approach was motivated by increasing evidence that source memory is enhanced for boundary representations (e.g., Heusser et al., 2018; Siefke et al., 2019), suggesting that attention and memory is enhanced for information following a context shift - perhaps reflecting the prioritization of novel incoming information representing the new environment (a new ‘event model’ of the world; Zacks et al., 2007; Zacks and Sargent, 2010).

For completeness, however, we have performed these correlations for your consideration. When we examined this specific 1.5s PCA time-window, the results showed the same patterns as the results reported in the 3-s PCA.

Across experiments, boundary-related loadings on component #4 remained significantly correlated with boundary-related time expansion effects in memory ($\rho = .25$, $p = .044$). This relationship showed a statistical trend effect in Experiment 3 ($\rho = .30$, $p = .11$) and Experiment 2 ($\rho = .25$, $p = .15$). We also found that, across experiments, boundary-related loadings on component #3 was still not significantly related to with temporal order impairment effects in memory ($\rho = -.13$, $p = .32$). This relationship was significant in Experiment 3 ($\rho = -.46$, $p = .011$) but not in Experiment 2 ($\rho = .18$, $p = .29$).

Comment 6: *More generally, it is unclear why the authors' primary focus is on arousal rather than on particular mechanisms, such as motor re-mapping, prediction error signaling, reorientation of attention, updating, etc., that are affected by contextual changes and presumably involved in event segmentation. The PCA approach adopted here to examine the pupil data seems well-suited for such purposes, yet the authors make inferences in terms of the broader concept of arousal. It is not apparent why arousal dynamics need to be interpreted as the cause of event segmentation and later memory phenomena, rather than being sensitive to and interacting with mechanisms such as the ones mentioned above. The arousal signals could, of course, be the cause, but then the experimental approach would need revision to be able to establish that. Currently, it seems more appropriate to conclude that the pupil tracks information processing influenced by event boundaries and that dissociable pupil components may reveal the contribution of distinct mechanisms with potentially different relevance for later subjective temporal memory. PCA decomposition seems promising to establish further which these mechanisms are and to specify their relative contributions.*

RESPONSE: We strongly agree with Reviewer 2 that an array of cognitive process known to be signaled by arousal and pupil dilation may contribute to event segmentation in memory. Throughout the revised manuscript, we now use language that pupil dilation tracks or signals arousal/cognitive processes that relate to and/or directly contribute to event segmentation in memory. For additional detail, please see our response to Reviewer 2's Comment #1.

****REVIEWERS' COMMENTS:**

Reviewer #2 (Remarks to the Author):

The authors have been responsive to the reviewer comments and made substantial changes to their manuscript, including additional analyses and rewriting, over the two revision rounds. While these changes have undoubtedly improved the manuscript, they have not ultimately eliminated the primary concern in my initial review; that is, clarified the functional significance of the reported pupil-memory correlations (Fig. 7). That would appear relevant as these constitute the main novel findings of the paper. Thus, although these results are interesting, the theoretical contribution of the paper remains somewhat unclear. Crucially though, the authors have in their second revision made numerous changes throughout the manuscript to eliminate unwarranted interpretations of their findings in terms of causality (arousal-> memory). I believe those changes were necessary.

I further believe that I have now had the opportunity to express all my concerns, and I have nothing further to add. Again, this is a very interesting study, with results that may prove important, and perhaps it is time for the readership to assess its theoretical contribution.